# Protein Dielectrophoresis: A Tale of Two Clausius-Mossottis—Or Something Else?

**DOI:** 10.3390/mi13020261

**Published:** 2022-02-06

**Authors:** Ronald Pethig

**Affiliations:** Institute for Integrated Micro and Nano Systems, School of Engineering & Electronics, The University of Edinburgh, The King’s Buildings, Edinburgh EH9 3JF, UK; ron.pethig@ed.ac.uk

**Keywords:** Clausius–Mossotti function, dielectric beta-dispersion, dielectrophoresis, electrokinetics, Lorentz cavity, Maxwell cavity, molecular dynamics simulations, proteins

## Abstract

Standard DEP theory, based on the Clausius–Mossotti (CM) factor derived from solving the boundary-value problem of macroscopic electrostatics, fails to describe the dielectrophoresis (DEP) data obtained for 22 different globular proteins over the past three decades. The calculated DEP force appears far too small to overcome the dispersive forces associated with Brownian motion. An empirical theory, employing the equivalent of a molecular version of the macroscopic CM-factor, predicts a protein’s DEP response from the magnitude of the dielectric *β*-dispersion produced by its relaxing permanent dipole moment. A new theory, supported by molecular dynamics simulations, replaces the macroscopic boundary-value problem with calculation of the cross-correlation between the protein and water dipoles of its hydration shell. The empirical and formal theory predicts a positive DEP response for protein molecules up to MHz frequencies, a result consistently reported by electrode-based (eDEP) experiments. However, insulator-based (iDEP) experiments have reported negative DEP responses. This could result from crystallization or aggregation of the proteins (for which standard DEP theory predicts negative DEP) or the dominating influences of electrothermal and other electrokinetic (some non-linear) forces now being considered in iDEP theory.

## 1. Introduction

The subject of protein dielectrophoresis (DEP) is at an important stage where a maturing theory, supported by molecular dynamics (MD) simulations of solvated proteins [1] has clarified aspects that have largely remained unresolved since the pioneering studies reported in 1994 by Washizu et al. [2]. In particular, the applied electric fields and field gradients are in many cases far too weak to generate DEP forces capable of overcoming the thermal (Brownian) force acting on a protein molecule [3,4,5]. There have also been inconsistencies regarding the polarity (positive or negative) of the observed DEP response for the protein most commonly studied (bovine serum albumin) [4]. These aspects have implications for the general field of molecular DEP. Are there generic lessons to be learnt from the evolving theory for proteins that can be applied to the DEP of small DNA fragments or ribosomal RNA, for example? Or perhaps each class of biomolecule will present its own particular challenge? Based on an empirical theory, it was proposed that the DEP response (including the DEP cross-over) of a globular protein can be predicted from the magnitude and frequency profile of its dielectric *β*-dispersion, which reflects the protein’s squared dipole moment and its relaxation time [4]. Support for this proposal is given by MD simulations for lysozyme and ubiquitin, which show that the *β*-dispersion also encompasses cross-correlations of the protein dipole with its hydration shell [1]. The corresponding protein–water Kirkwood correlation factor is found to be close to unity. This implies little correlation, in line with an earlier assumption for myoglobin [6], but is impossible to prove using conventional dielectric spectroscopy.

Figure 1 serves to identify the two ‘Clausius–Mossottis’, [*CM*]*_macro_* and [*CM*]*_molecular_*, of this article’s title and to indicate that the new theory holds the key for them both [1]. This figure also prompts clarification of the assertion that [*CM*]*_macro_* should not be employed to analyze the DEP responses of proteins and other biomolecular particles that possess a permanent dipole moment [3]. On account of the Janus nature of the DEP susceptibility factor derived by Heyden and Matyushov for proteins [1], this question is relevant and can be answered as follows.

The word ‘dielectrophoresis’ (DEP) implies a particle being carried (i.e., moved) by a dielectric force. As taught in early texts, this is referred to as the action of a ponderomotive force (able to move an object of mass) and is defined in terms of the gradient of the particle’s potential energy *U* when placed in an electric field gradient [11,12]:(1)$$F_{P}=-\frac{\partial \left(U-U_{0}\right)}{\partial r}=-\nabla U_{P}$$

The force F*_p_* thus drives the particle towards places of field strength where the particle’s free energy (chemical potential) is reduced. Although this relationship can be used to analyse the DEP of a particle possessing both an induced and a permanent dipole moment, standard DEP theory defines this in terms of the force acting on its moment *m* in an electric field gradient $\nabla E_{m}$ [13,14]:(2)$$F_{DEP}=m\cdot \frac{\partial E_{m}}{\partial r}=\left(m\cdot \nabla \right)E_{m}$$

Equations (1) and (2) are equivalent because the potential energy of a *rigid* dipole (i.e., unable to rotate) is given by the following relationship [15]:(3)$$U=-\left(m\cdot E_{m}\right)$$

This equivalence provided by Equation (3) is restricted to where the dipole’s polar angle of orientation *θ* with respect to the applied field remains fixed. A change of *θ* results in a change of *U* and produces a restoring torque T=−∂U∂θ that acts on the moment. The work done (i.e., energy gained) is the product of force applied over a distance, so that for a rigid dipole Equations (1) and (2) are connected through the relationship $U_{DEP}=-\int F_{DEP}.$ This predicts that if the electric polarizability of the particle *exceeds* that of the fluid medium it has displaced, it will minimize its potential energy by seeking a region of *high* field strength near an electrode. Work must be done on the particle to move it away, against the field, from an electrode (i.e., the field does negative work). By convention this is termed *positive* DEP. Negative DEP corresponds to where a particle of lower polarizability than its surrounding medium moves to a local field minimum and away from the electrodes. In practical applications, such as the selective separation of a target particle from a mixture, it is important to know how the DEP force depends on the frequency of the applied field. In standard DEP theory for macroscopic particles (e.g., cells, bacteria or polymer beads) this information is carried by the macroscopic Clausius–Mossotti factor [*CM*]*_macro_*. This theory has worked well for DEP studies that have progressed from the micron-scale of mammalian cells, microalgae and bacteria, down to the submicron-scale of virions and small vesicles [16]. These particles do not carry a permanent dipole moment. Replacing the moment *m* of Equation (2) with the known value of the permanent dipole moment of a protein does not solve the problem as to how the DEP force can overcome the energy *kT* of thermal ‘noise’ and Brownian motion (Boltzmann constant *k* and absolute temperature *T*) [4]. The aspect absent from Equations (2) and (3) is the fact that globular proteins are usually free to ‘tumble’ in a medium such as water. It is now clear that an essential aspect of DEP theory, for particles possessing a permanent dipole moment that can rotate, is to replace the rigid dipole moment *m* of Equation (2) with its time-averaged, field-oriented moment proportional to (*m*^2^/3*kT*)E*_m_* [1].

Pohl in his research articles and book [13] does not adopt the [*CM*] factor. Herman Schwan, a pioneer of electrical bioimpedance, is probably responsible for its use in electrorotation theory, where he refers to it as ‘*effectively a macroscopic application of the Clausius–Mossotti factor*’ [17]. Later, Jones [14] adopts the ‘*Clausius–Mossotti function*’ in the theory for DEP. However, this practice is not to be confused with application of the Clausius–Mossotti law [10]—identified in Figure 1 as [*CM*]*_molecular_*. This law relates the relative permittivity of a dielectric material to the number density of polarizable elements in its molecular structure—of which tumbling molecular dipoles are particularly important elements [18,19]. The genesis of [*CM*]*_molecular_* differs significantly from that of [*CM*]*_macro_*.

The theoretical foundations of [CM]*_macro_* and [CM]*_molecular_*, are Green’s theoretical formulation of electrical potential [7], Faraday’s discovery of specific inductive capacity (i.e., relative permittivity *ε**_r_*) [8], and Mossotti’s hypothesis that the electric fluid residing in a “corpuscle” within a dielectric is displaced under the action of a local field to form an electric doublet (i.e., induced dipole) [9]. Green derived the electric force exerted by a charged spheroid (electrical density *ρ*), on a volume element at a distant point *r*, in terms of a potential function V=∭ρrdυ. Outside the spheroid $\nabla^{2}V=0$ (Laplace’s equation) and inside it $\nabla^{2}V=-\rho /\varepsilon_{0}$ (Poisson’s equation), given here in SI units with *ε*_0_ the permittivity of vacuum. Clausius used these concepts to derive his eponymous factor [*CM*]*_molecular_* [10]. Mossotti’s hypothesis influenced Maxwell’s concept of the displacement current density, D = *ε*_o_*ε_r_*E*_m_* [20]. The derivation of [*CM*]*_macro_* is obtained by solving Laplace’s equation [3], using boundary rules for Green’s potential function (it must be continuous) and for Maxwell’s displacement current density (its normal component must be continuous) at the interface between different dielectric materials. An important innovation of the new theory is to identify that, for protein DEP theory, these boundary rules of macroscopic electrostatics should be replaced by calculation of cross-correlations of the protein dipole with water dipoles in its hydration shell [1]. The new theory also takes into account the polarization of the protein’s hydration shell induced by the protein’s permanent dipole moment. These should be considered as generic aspects for molecular DEP, and highlight the importance of dielectric spectroscopy and MD simulations going forward.

Finally, aspects of ‘Something Else’ included in this article’s title address experimental issues such as control of protein and suspending medium stability, together with determinations of the magnitudes of the applied field and field gradient. Experimental guidelines for protein DEP were set by Washizu et al. [2], such as an upper limit for protein concentration (0.1 μg/mL) and solvent conductivity (1 mS/m). Researchers engaged in eDEP, where conductive electrodes generate the required field gradient, have tended to follow these guidelines. In such cases, there are consistent reports of a positive DEP response observed for the 22 different proteins so far studied [4], in line with the new theory plus MD simulations [1] and as predicted by the empirical theory [4]. Experimenters who employ insulating structures to create the field gradients (iDEP) have sometimes stretched these experimental guidelines to explore new opportunities for exploiting protein DEP. For example, protein concentrations of 1 gm/mL and solvent conductivities of 100 mS/m have been used. In such cases, negative rather than positive protein DEP has been reported (e.g. [21,22,23]). As indicated in Figure 1, and discussed in Section 2, the theory underpinning [*CM*]*_macro_* employs the Maxwell cavity field, with the limitation of a *minimum* cavity size *below* which the rules of classical electrostatics theory may not hold. For the Lorentz cavity field employed in the derivation of [*CM*]*_molecular_*, there is a *maximum* cavity size *above* which cavity field theory breaks down. This interesting situation [24] could provide an experimental approach to determining and controlling a threshold for the particle size above which protein DEP is governed by [*CM*]*_macro_* rather than [*CM*]*_molecular_*. For example, in an assessment of future protein-based drug carriers, standard DEP theory correctly predicts the reported separation by negative DEP of *micron*-sized designer protein particles according to their size and shape [25]. Whilst, for BSA assumed to be in its monomolecular form, *positive* iDEP is observed at a concentration of 7 nM [26,27] and *negative* iDEP at a concentration of 0.15 mM [21]. As discussed in Section 5, these conflicting results were obtained using direct current (DC) iDEP experiments, where non-linear electrophoretic and electroosmosis effects can overcome DEP. Finally, the phenomenon of pearl chaining, where particles aggregate under the influence of an applied field, was identified and experimentally avoided as a possible confounding issue for molecular DEP by Washizu et al. [2]. The possibility that field-driven protein aggregation can occur at a threshold of protein concentration, which may provide new opportunities of exploitation, also merits further consideration under the category of ‘Something Else’.

## 2. Limitations of Macroscopic Electrostatics with Respect to Protein DEP

### 2.1. The Maxwell Cavity Susceptibility and [CM]_macro_

[*CM*]*_macro_* is derived from calculation of the field inside a particle (relative permittivity *ε**_p_*) residing within a dielectric medium (relative permittivity *ε**_m_*) in which a uniform field E*_m_* has been established. Charges are created at the dielectric discontinuity represented by the particle–medium interface, to produce an interface dipole moment, M*^int^*. The polarization of the particle, P*_p_*, expressed as the induced dipole moment per unit volume, is given by:(4)$$P_{p}=\frac{M^{int}}{V_{p}}=\left(\varepsilon_{m}-\varepsilon_{p}\right)\varepsilon_{0}E_{i}=\chi_{p}\varepsilon_{o}E_{i}$$
where *χ_p_* is termed the electric susceptibility of a volume of a material. As described in Section 3, dielectric theory often employs the susceptibility rather than the permittivity of a material. In standard DEP theory, the induced interface dipole moment, M*^int^*, of Equation (4) is the moment *m* required in Equation (2). Equation (4) is obtained by solving Laplace’s equation and involves applying the boundary rules mentioned in the Introduction. The field E*_i_* created within the particle is uniform and directed along the same axis as E*_m_* (but opposes it for *ε_p_* > *ε_m_*) and is given by [3,28]:(5)$$E_{i}=\frac{3\varepsilon_{m}}{2\varepsilon_{m}+\varepsilon_{p}}E_{m}$$

A central aim of molecular dielectrics is to understand the response of a single dipole moment to an external field. Maxwell’s approach is to carve out an evacuated cavity inside a dielectric [20]. The field inside an evacuated spherical cavity (i.e., *ε_p_* = 1) is known as the Maxwell cavity field. The susceptibility *χ_Mc_* of the Maxwell cavity is defined as the ratio of this field and the applied external field, so that from Equation (5):(6)$$\chi_{Mc}=\frac{3\varepsilon_{m}}{2\varepsilon_{m}+1}$$

The mathematical derivation of P*_p_* in Equation (4) is obtained by solving Laplace’s equation, which assumes that no free charges already exist at the interface. Macroscopic particles usually carry a net surface charge, but if uniformly distributed over their surface this charge does not contribute a component to be added to P*_p_* [3]. The time taken for M*^int^* to fully develop is typically around 100 nanoseconds, dictated by the relaxation time of the interfacial charges. This, in turn, depends on the permittivity and conductivity of the particle and surrounding medium, but does not depend on particle size [3,16]. For a sphere of radius *R*, then from Equations (4) and (5):(7)$$M^{int}=V_{p}P_{p}=3V_{p}\varepsilon_{0}\varepsilon_{m}\left[\frac{\varepsilon_{p}-\varepsilon_{m}}{\varepsilon_{p}+2\varepsilon_{m}}\right]E_{m}=4\pi R^{3}\varepsilon_{0}\varepsilon_{m}{\left[CM\right]}_{macro}E_{m}$$

The term in square brackets, [*CM*]*_macro_*, is limited to the range −0.5 < [CM]*_macro_* < 1.0. To accommodate the phase difference between dielectric displacement and ohmic conduction currents, complex dielectric parameters (i.e., with real and imaginary components) are used in this definition. This modification of Maxwell’s original DC model is known as Maxwell–Wagner polarization, with the real component of [*CM*]*_macro_* required in Equation (7) [13,14,16]. Displacement currents tend to dominate above ~50 kHz, so that the relative permittivity values, *ε*_p_ and *ε*_m,_ are used directly in Equation (7), to be replaced with conductivity parameters *σ_p_* and *σ_m_* below ~1 kHz.

Equation (7) describes a spherical particle, but ~75% of globular proteins take the form of prolate spheroids, with ~25% being oblate [29]. The following general result for M*^int^* aligned along an axis parallel to the direction of E*_m_* can be applied [15]:(8)$$M^{int}=\frac{4\pi abc}{3}\varepsilon_{0}\varepsilon_{m}\left[\frac{\varepsilon_{p}-\varepsilon_{m}}{\varepsilon_{m}+A\left(\varepsilon_{p}-\varepsilon_{m}\right)}\right]E_{m}$$
where *a, b, c*, are the spheroid’s semiaxes. The factor *A* quantifies the shielding of the particle’s interior from E*_m_* by the induced interfacial charges. For a sphere *A* = 1/3, whilst for a prolate spheroid tending towards the shape of a long thin cigar, with its axis parallel to the external field, *A* tends to zero (i.e., the internal and external fields are similar). For an oblate spheroid tending towards a thin platelet, with its major axis perpendicular to the field, *A* approaches 1.0 (the internal field is reduced by the factor *ε_p_*/*ε_m_*).

In modeling the polarization of spheroids, there are advantages to retaining volume *V_p_* of Equation (7) as an independent parameter [30]. The DEP force can then be defined in terms of a susceptibility factor *χ_DEP_* that scales linearly with particle volume [1]:(9)$$F_{DEP}=\varepsilon_{0}\chi_{DEP}\nabla E_{m}^{2}$$

For a spherical particle (*a* = *b* = *c* = *R*), application of Equations (2) and (6), together with the vector transformation $2\left(E\cdot \nabla \right)E=\nabla E^{2}$, gives:(10)$$\chi_{DEP}=\frac{3}{2}V_{p}\varepsilon_{m}\left\{Re\left[\frac{\varepsilon_{p}^{*}-\varepsilon_{m}^{*}}{\varepsilon_{p}^{*}+2\varepsilon_{m}^{*}}\right]\right\}=\frac{3}{2}\;V_{p}\varepsilon_{m}\;Re{\left[CM\right]}_{macro}^{*}$$
where the asterisks indicate complex permittivity values and *Re* the real component. An important application of DEP is the selective separation of a target particle from a mixture based on its characteristic ‘cross-over’ frequency *f**_xo_* (Hz). The frequency-dependence of the Clausius–Mossotti factor in a range that encompasses *f**_xo_* is given by [16]:(11)$$Re{\left[CM\left(f\right)\right]}_{macro}^{*}\approx \frac{f^{2}-f_{xo}^{2}}{f^{2}+2f_{xo}^{2}}$$

For frequencies of the applied field less than *f**_xo_* the DEP force acting on the particle is negative. As the frequency *f* is increased above *f**_xo_*, a transition from negative to positive DEP occurs. Different proteins exhibit different values of *f**_xo_* [16,24], so there is the expectation that this procedure, already proven for the selective separation and manipulation of cells and bacteria, for example, can be used to selectively enrich target proteins from other proteins or biomacromolecules.

Based on common aspect ratios for prolate and oblate proteins [29], the depolarization factor of relevance has limits 0.2 < *A* < 0.8. Provided that the protein sample is large enough for it to be characterized using macroscopic electrostatics then, from Equation (10), the corresponding DEP susceptibilities related to their induced moments are:(12)$$\chi_{DEP}=\frac{3}{2}V_{p}\varepsilon_{m}\left[\frac{5(\varepsilon_{p}-\varepsilon_{m})}{\varepsilon_{p}+4\varepsilon_{m}}\right]\;\;\;\;\;(\mathrm{prolate}\;\mathrm{spheroid}:\;A=0.2)\;\chi_{DEP}=\frac{3}{2}V_{p}\varepsilon_{m}\left[\frac{5(\varepsilon_{p}-\varepsilon_{m})}{4\varepsilon_{p}+\varepsilon_{m}}\right]\;\;\;\;\;(\mathrm{oblate}\;\mathrm{spheroid}:\;A=0.8)$$

These relationships indicate that the free energy per unit volume -(M·E*_m_*), where M is the vector sum of its constituent dipole moments, depends on the shape of a macroscopic particle. This indicates that, even for macroscopic distances, interactions between the constituent dipole moment fields should be taken into account. However, an accurate calculation of the interaction of a specific dipole with all the other dipoles in the sample would be difficult to achieve. The common procedure in dielectrics theory [19] is thus to consider the dipole in question to be at the centre of a sphere containing a finite number of other dipoles, beyond which there is a homogeneous continuum of the dielectric medium. For this model to represent a good approximation, the quantity of the other dipoles in the spherical volume should be sufficient that the ‘law of large numbers’ [28] is applied to ‘smooth-out’ the influence of random polarizations and so lend to the sphere an effective permittivity equal to that of a macroscopic sample. This is of relevance to the dielectric models discussed in Section 3.

To overcome Brownian disruptive effects, the DEP force should exceed the thermal energy (3*kT*/2) of the protein’s translational degrees of freedom. From Equation (9) the threshold value of $\nabla E_{Th}^{2},\;$ above which F*_DEP_* can overcome thermal disruption, is given by:(13)$$\nabla E_{Th}^{2}=\frac{3kT}{2\varepsilon_{0}\chi_{DEP}}$$

Threshold values for $\nabla E_{Th}^{2}$ are shown in Figure 2 for *T* = 295 K as a function of effective particle radius, employing values for *χ_DEP_* given by Equation (12) with *ε_p_* = 25, *ε_m_* = 78. The value *ε_p_* = 25 is obtained from a molecular dynamics (MD) simulation of cytochrome-c in a water droplet of 2.4 nm radius [31], and considered to be consistent with hydration-dependent dielectric properties of protein powders [32]. Also shown in Figure 2 are values of the factor $\nabla E_{m}^{2}$ reported to have been used in protein DEP experiments. It appears [4] that for only two cases [22,33], involving bovine serum albumin (BSA) and immunoglobulin G (IgG), does the experimental magnitude of $\nabla E_{m}^{2}$ exceed the threshold required to overcome thermal randomization. For two cases involving the prolate proteins BSA and IgG [34,35] the threshold is close to being met. For the other cases [26,27,36,37,38,39] shown in Figure 2, experimental values of the field gradient factor $\;\nabla E_{m}^{2}$ fall orders of magnitude below that required (according to standard theory) to observe DEP if proteins of monomolecular form had been studied.

As discussed in Section 3, solvated proteins are free to tumble and possess an orientational polarizability, $m_{p}^{2}/3kT$*,* associated with its permanent dipole moment *m**_p_* [18]. From Equations (2) and (9) the DEP susceptibility, *χ_dDEP_*, associated with such a dipole is given as:(14)$$\chi_{dDEP}=\frac{F_{DEP}}{\varepsilon_{o}\nabla E_{m}^{2}}=\frac{1}{\varepsilon_{o}\nabla E_{m}^{2}}\left(\frac{m_{p}^{2}}{3kT}E_{m}\cdot \nabla \right)E_{m}=\frac{m_{p}^{2}}{6\varepsilon_{o}kT}$$

The following gives the contribution to the DEP force of a rigid dipole moment relative to that of an induced moment:(15)$$\frac{F_{dDEP}\left(rigid\;dipole\right)}{F_{DEP}\left(induced\right)}=\frac{m^{2}}{6kT\varepsilon_{0}\chi_{DEP}}$$

As shown in Figure 3, standard DEP theory based on Equation (2) predicts that a rigid dipole moment will experience roughly the same DEP force as an induced moment under the same experimental conditions. This confirms an earlier conclusion [4], based on an analysis only for BSA, that the problem regarding the apparent absence of Brownian disruption for protein DEP is not solved by inserting the magnitude of the permanent dipole moment value into Equation (2). A DEP theory based on molecular dielectrics is required—not one based on macroscopic electrostatics. 

Aqueous solutions of proteins, when examined by dielectric or impedance spectroscopy at room temperature, typically exhibit a frequency-dependent relative permittivity as shown in Figure 4. Of particular significance for protein DEP is the *β*-dispersion, bounded by lower and upper frequency permittivity values of ε(βlf) and ε(βhf), respectively, with a dispersion strength given by Δε(*β*) = [ε(βlf) − ε(βhf)]. The mid-point, [ε(βlf) − ε(βhf)]/2, of this dispersion occurs at a frequency 1/(2*π**τ*), where *τ* is the characteristic relaxation time. This is the time required for 1/*e* of the permanent dipoles of an ensemble to return to random orientation after the polarizing field has been removed. For suspensions of cells and other particles of sufficiently low volume concentration, *c_v_*, the magnitude of Δε(*β*) is given by [3,4,16]:(16)$$\Delta \varepsilon \left(\beta \right)=3c_{v}\varepsilon_{m}{\left[CM\right]}_{macro}$$

However, as indicated in Figure 4, whereas the *δ*-dispersion associated with a protein’s hydration shell appears to satisfy this relationship, the *β*-dispersion does not. Equation (16) is derived from the ‘effective medium’ theory of dielectric mixtures [24]. This theory assumes that the addition of a small number of ‘impurity’ particles to a homogeneous dielectric medium can be homogenized, so that after this process the mixture exhibits the same polarization response as the pure solvent. In effect, the ratio of the average electric displacement and average electric field 〈D⟩/〈E⟩ is assumed to remain unchanged. The fact that the theory fails for the *β*-dispersion exhibited by a protein–water mixture implies that it is not a ‘passive’ mixture—an interaction occurs between the protein and water dipoles. The magnitude of the *γ*-dispersion observed at ~20 GHz, shown in Figure 4 and due to orientational relaxation of the water dipoles, was understood after introduction into dielectrics theory of the Kirkwood water dipole–dipole correlation factor. An obvious matter to raise is the extent to which the magnitude of the *β*-dispersion might be influenced by protein–water cross-correlations of dipoles. As discussed here in Section 3 and Section 4, this concept had evaded analysis with standard dielectrics theory, but is an important component of the new theory and MD simulations [1].

### 2.2. The β- and δ-Dispersions Associated with Protein Hydration

It is the custom to identify a dielectric dispersion, using the Greek alphabet, in the order of location on the frequency axis of the applied field frequency. The designation of a particular Greek letter does not specify a specific physico-chemical mechanism responsible for that dispersion. Figure 5 shows the *β*-dispersion exhibited by solvated BSA [40,41], together with an aqueous suspension of phospholipid nano-vesicles [42].

The dispersion for the vesicles is characterized by a relaxation time *τ* corresponding to that required for 1/*e* of the total induced interfacial charges to disappear after the polarizing field has been removed. The magnitude of the *β*-dispersion, as a function of vesicle volume fraction, *p*, is given by the linear relationship Δε(*β*) ≈ 143*p* [42]. From Equation (16) this gives [*CM*]*_macro_* = 0.61 [42], a value within the permitted range −0.5 < [*CM*]*_macro_* <1.0. However, an equivalent interpretation is not forthcoming for the *β*-dispersion shown in Figure 5 for a 12 g/L (0.18 mM) BSA solution [40]. Based on its molecular mass of 66 kD and mass density 1.41 g/cm^3^ [43], BSA has a molecular volume of 4.68 × 10^4^ cm^3^/mol. This corresponds to the protein occupying 8.42 cm^3^/L in a 12 g/L mixture, so that *p* = 8.4 × 10^−3^. Thus, the *β*-dispersion for BSA exhibits a significantly larger dispersion strength than the vesicles, with Δε(*β*) ≈ 2500*p* and a corresponding value for [*CM*]*_macro_* well above the maximum of 1.0 permitted by the standard theory.

**Figure 5 micromachines-13-00261-f005:**
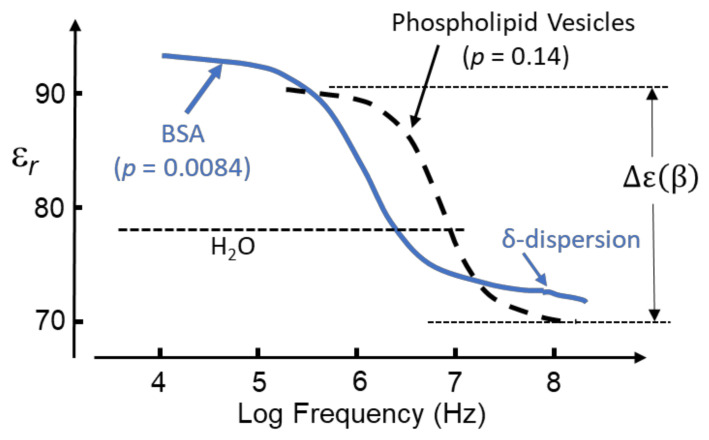
The *β*-dispersion exhibited by a 0.18 mM (i.e., *p* = 0.0084) BSA solution [40] is shown, together with that of a suspension of phospholipid vesicles (radius 13.5 nm, volume fraction *p* = 0.14) [42]. The *δ*-dispersion is associated with the protein hydration sheath [41,44,45]. Positive DEP can be expected for both particle types in the frequency range where their relative permittivity, *ε_r_*, exceeds that (~78) of pure bulk water.

The *δ*-dispersion, typically occurring around 70 MHz, is attributed to relaxations of water molecules close to the protein surface [41,44,45]. Two kinds of such water have been identified: (i) water attached directly to the protein surface via H-bonding to oxygen, nitrogen and polar groups; (ii) water, amounting to ~0.36 g water/g protein, that forms a hydration shell around the protein molecule. The magnitude of the *δ*-dispersion, Δε(*δ*), is found to be directly proportional to the molecular weight and hence surface area of the protein molecule, whereas the value of its characteristic relaxation time (~2 ns) is relatively independent of protein size [45]. The number of bound water molecules can be estimated from Δε(*δ*) using a simple mixture theory [16,44]. For small proteins, such as cytochrome-c or ribonuclease-A, the number of bound water molecules amounts to ~50 per protein, increasing to 135~150 for larger proteins such as hemoglobin and BSA [44]. Based on measurements of the *β*-, *δ*- and *γ*-dispersions exhibited by lysozyme solutions, Wolf et al. conclude that the dynamics of water molecules in the hydration sheath is influenced by interactions with polar residues on the protein surface and less so with the bulk water [45]. This implies that the *δ*-relaxation is strongly coupled to the *β*-relaxation of the protein molecule and its H-bonded water molecules but is not strongly influenced by structural fluctuations of the bulk solution (i.e., the *γ*-dispersion). It is also noteworthy, as indicated in Figure 4, that the magnitude of the dielectric increment Δε(*δ*) corresponds to a realistic value for [*CM*]*_macro_*—unlike the case for ∆ε(*β*). For example, a 5 wt.% aqueous solution of myoglobin (equivalent to *p* = 0.034) has a measured Δε(*δ*) = 2.37 at 25 °C [44]. From Equation (16) this gives [*CM*]*_macro_* = 0.3, which lies within the permitted range −0.5 < [*CM*]*_macro_* <1.0. This conclusion also holds for nine other globular proteins of similar concentration investigated by Miura et al., who reported Δε(*δ*) values in the range 1.51 to 5.05 [44]. These dispersion strengths correspond to [*CM*]*_macro_* values in the range 0.19 to 0.63. This implied compliance with classical macroscopic electrostatics is matched by the characteristic relaxation time being independent of protein size. As stated with respect to Equation (5), the relaxation time of induced interfacial charges associated with the Maxwell cavity field depends on the permittivity and conductivity of the particle and surrounding medium, but not on particle size [14,45].

### 2.3. User-Friendly, Empirical, Theory for Protein DEP

A practical and user-friendly application of Equation (16) arises because Δ*ε*(*β*) contains information for predicting the frequency profile of the DEP response, and where a transition from positive to negative DEP might be expected. For proteins whose *β*-dispersion is well characterised, and for a protein concentration in the range where a linear relationship between Δ*ɛ*(*β*) and *C**_v_* is found, the variation of permittivity of the aqueous mixture as a function of frequency (Hz) is of the form [18]:(17)$$\varepsilon_{m}\left(f\right)=\frac{\Delta \varepsilon \left(\beta \right)}{1+{\left(f/f_{\beta }\right)}^{2}}+\varepsilon \left(\beta_{hf}\right)$$
where *f_β_* is the frequency marking the inflexion (mid-point) of the *β*-dispersion and *ε(β_hf_*) is the high-frequency boundary where the dispersion ends. The dispersion exhibits a dielectric increment in the frequency range where *ε(β_hf_*) exceeds the experimental value of 78.4 for pure water [3]. This corresponds to where the polarizability (dipole moment per unit volume) of the solvated protein *exceeds* that of the medium it has displaced. The transition to the dielectric decrement occurs at the cross-over frequency, *f**_xo_*, where *ε**_m_* falls below 78.4. The polarizability of the solvated protein is now *less* than that of the medium it has displaced, and a transition from positive to negative DEP can be expected. For the BSA dispersion of Figure 5: Δ*ɛ*(*β*) = 21; *ε(βhf*) = 72.4; *f**_β_* ≈ 1 MHz; *f**_xo_* ≈ 2 MHz.

Equation (16) is derived from the ‘effective medium’ theory of dielectric mixtures. For a mixture of protein and water this theory can be applied to analyse the *δ*-dispersion of Figure 4, but fails for the *β*-dispersion. This theory assumes that, after the addition of a small number of impurity particles, the mixture can be homogenized so that it exhibits the same polarization response as the pure solution. In effect, the ratio of the average electric displacement and average electric field 〈D⟩/〈E⟩ is assumed to remain unchanged. For particles of permittivity *ε_p_* and total partial volume *v_p_* dispersed in a medium of permittivity *ε_m_*, the following relationship is derived [46]:(18)$$\left(\varepsilon_{eff}-\varepsilon_{p}\right)v_{p}\frac{\langle E_{i}\rangle }{E_{m}}+\left(\varepsilon_{eff}-\varepsilon_{m}\right)(1-v_{p})\frac{\langle E_{m}\rangle }{E_{m}}=0$$
where $\langle E_{i}\rangle$ and $\langle E_{m}\rangle$ represent the statistical averages of the electrical field in the interior of a dispersed particle and in the bulk medium, respectively. The field E*_i_* inside each particle is taken to be the Maxwell cavity field given by Equation (5), and substitution of this into Equation (16) gives:(19)$$\frac{\varepsilon_{eff}-\varepsilon_{m}}{\varepsilon_{eff}+2\varepsilon_{m}}=v_{p}\frac{\varepsilon_{p}-\varepsilon_{m}}{\varepsilon_{p}+2\varepsilon_{m}}=\nu_{p}{\left[CM\right]}_{macro}$$

This Equation provides a valuable tool, known as multi-shell modelling, to predict and understand the DEP responses of cells and bacteria [16,47]. It was used by Schwan et al. [42] to derive the lipid bilayer capacitance of the phospholipid vesicles from its *β*-dispersion shown in Figure 5, and to analyse the *δ*-dispersion exhibited by solvated BSA [41]. Equation (16) is derived from Equation (19) by assigning (*ε_eff_* − *ε_m_*) = ∆*ε*(*β*), *v_p_* = *p*, together with the assumption that *v_p_* is small enough to give *ε_eff_* +2*ε_m_* ≈ 3*ε_m_*. An empirical replacement for [*CM*]*_molecular_* of Figure 1, namely ${\left[CM\right]}^{empirical}$, is derived by assigning *v_p_* as a mole volume fraction *C**_p_* and inserting *ε_eff_* = *κε_m_* into the denominator of the left-hand side of Equation (18) [4]:(20)κ+2CMempirical=ΔεβεmCwρpCpρw

Based on known values for ∆*ε*(*β*)/*C**_p_* and protein mass density [4] this relationship produces values for ${\left[CM\right]}^{empirical}$ that vary greatly in magnitude (e.g., ~1110 for BSA; ~37,000 for carboxypeptidase; ~190 for phospholipase) with no apparent dependence on protein molecular weight. Selected examples are given in Figure 6.

If the values for *f**_xo_* in Figure 6 indicate where transition from positive to negative DEP occurs, a protocol for separating a mixture of cytochrome-c and the other proteins can be devised. By setting the DEP voltage frequency at ~15 MHz, cytochrome and ubiquitin would be attracted to an electrode array by +ve DEP, with the other proteins repelled by -ve DEP into the (flowing) bulk medium. By adjusting the frequency to ~30 MHz, separation of cytochrome and ubiquitin could result.

The DEP susceptibility profiles shown in Figure 6 also provide a possible explanation for DEP cross-over events reported in the MHz range for BSA [33], avidin [48] and prostate specific antigen [49]. This data is shown in Figure 7.

The generic expression for the DEP susceptibility is given as
(21)$$\chi_{DEP}=\frac{3}{2}\;V_{p}\varepsilon_{m}\;\left(\kappa +2\right){\left[CM\right]}^{empirical}$$

The parameter *κ* can assume two values, either *κ* = 1 or *κ* >> 1. The case *κ* = 1 is given where the particle concerned does not possess a permanent dipole moment, so that *χ_DEP_* is given by Equation (10). The *β*-dispersion it exhibits, when suspended in a fluid, is capable of being analysed in terms of Equation (16) and the DEP force it experiences is given by Equation (9). For protein DEP and other cases of molecular DEP where the sample macromolecule possesses a permanent dipole moment, then *χ_DEP_* is given by:(22)χDEP=32Vpεmκ+2CMempirical=32 VpεmΔεβεmCwρpCpρw

The frequency dependence of the DEP force now assumes a modified version of Equation (9), based on Equation (17):(23)FDEPf=32 Vpε0εmfΔεβεmCwρpCpρw∇Em2

The dielectric relaxation time, *τ*, for a macromolecule is proportional to its volume [16] and so we can expect the *β*-dispersion for a protein dimer to be shifted to a frequency that is lower than its monomer counterpart, and also possibly exhibit a larger dispersion strength because of an increased effective dipole moment. Moser et al. [40] investigated the *β*-dispersion exhibited by monomer and dimer BSA. The characteristic frequency, *f_β_*, of Equation (17) was determined to be ~400 kHz at 25 °C for the dimer, compared to ~1 MHz for the monomer. The dispersion strength Δ*ε*(*β*) exhibited by the dimer was also determined to be ~14% larger than that for the monomer. This offers the possibility that DEP can be used to separate the polymeric forms of a protein.

## 3. Summary of Dielectric Theory of Relevance to Protein DEP

Although the term ‘dielectrophoresis’ implies that dielectric theory already underpins the theory and practice of DEP, the extent of this is actually limited to narrow aspects of macroscopic electrostatics. New concepts for molecular DEP, and for protein DEP in particular, have been introduced by Heyden and Matyushov [1], and they highlight the importance of MD simulations as a complimentary tool to dielectric spectroscopy. This summary is to help establish, primarily for members of the DEP community not trained in dielectrics, what is new to established theory and as a guide to the key literature.

Standard DEP theory lacks the ability to adequately deal with particles that possess a permanent dipole moment. The primitive depiction in Figure 8a of a globular protein molecule illustrates that significant contributions to its permanent dipole moment are its ionized acidic and basic peptides located on its surface. An estimate of the moment is calculated from the algebraic summation of the moment vectors, *q_i_r_i_*, directed towards the protein’s centre of mass. X-ray diffraction data provide the ‘resting’ locations of these charged groups. A more accurate value is obtained by including peptide moments and those formed by α-helixes [16]. A key innovation [1] is to replace the ‘boundary problem’ of macroscopic electrostatics with calculation of the cross-correlations of the protein dipole and neighbouring water dipoles that, in the inner hydration shell, are located close to the hydrophilic polar and charged side groups on the protein’s surface.

For the case shown in Figure 8b, dipole–dipole interactions enhance the effective dipole moment of the solvated protein. As explained in this section, this corresponds to a Kirkwood correlation factor, *g**_k_*, larger than unity. If anti-correlation occurs, then *g**_k_* is less than unity. The fluctuations of the moment shown in Figure 8c should fit a Gaussian distribution about its mean value. MD simulations for lysozyme and ubiquitin give mean dipole moment values of 145 D and 218 D, respectively, with Gaussian widths of 29 D and 37 D, respectively [1]. Thus, for these proteins, calculations that neglect modulations of the protein dipole moment by conformational fluctuations (i.e., assumes $M_{p}^{2}\approx \langle M_{p}^{2}\rangle$ does not introduce an error exceeding 4%.

The theory of linear response and susceptibility is important. The susceptibility χ(*ω*) of a system, when a sinusoidal force F of angular frequency *ω* is applied to it, is given as χ(*ω*) = R/F, where R is the response of the system. The interaction energy between F and R is equal to the negative dot product -F·R. This is the vector notation for -FR cos *θ*, where *θ* is the polar angle between F and R. In dielectrics theory, F is the applied field and R is the resulting dipole moment M_T_ of the system. The dielectric susceptibility of a material is equal to $\langle M_{T}^{2}\rangle {}_{0}/3kT$ where $\langle M_{T}^{2}\rangle {}_{0}$, as shown in Figure 8d, corresponds to the moment’s mean square fluctuation *without* application of the force field. Linear response theory connects the *production* of spontaneous thermal fluctuations of a system to the *dissipation* of excess energy produced by externally applied perturbation. Provided that a system of dipoles is at equilibrium when an external (relatively weak) electric field is applied to it, the temporal response of this system to the field can be determined in terms of $\langle M_{T}^{2}\rangle {}_{0}$. Applications of this theory extends to other topics, such as magnetism and elasticity. For example, by examining the spontaneous microscopic fluctuations of a spring-balance with no mass attached to it, a prediction can be made of the extent it will stretch if a specified mass is attached!

Figure 8 depicts important ingredients of protein DEP theory. What follows adds substance to this.

**Figure 8 micromachines-13-00261-f008:**
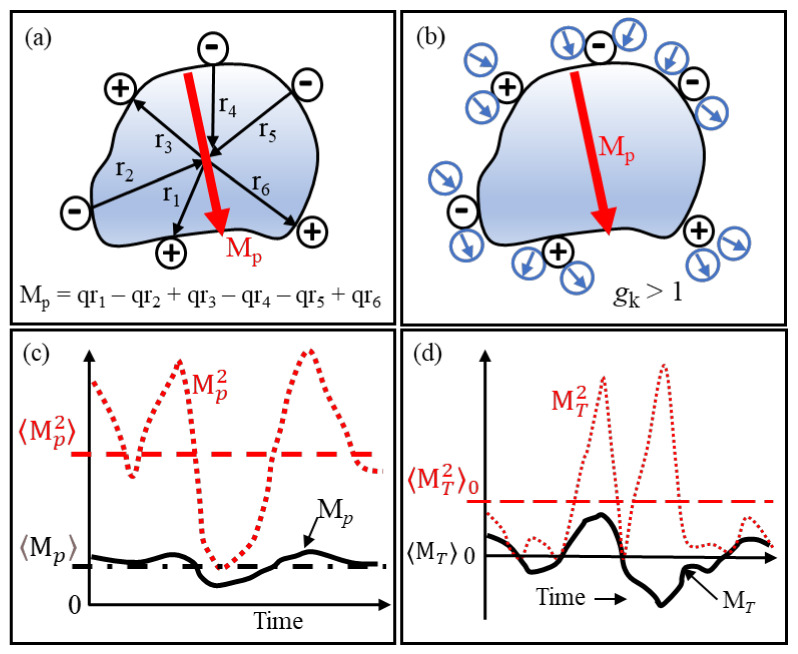
(**a**) Major contributions to the permanent dipole moment M_p_ of a protein are its peptide groups that carry a charge *q*. (**b**) H_2_0 dipoles near charged groups can act to enhance M_p_, corresponding to a correlation coefficient *g*_k_ > 1. Reversal of the polarities shown for the water dipoles gives anti-correlation, with *g**_k_* < 1. (**c**) Brownian (*kT*) changes of location and magnitude of *q* produce fluctuations with time of M_p_ about its mean value $\langle M_{p}\rangle$. The dashed red line indicates the mean square $\langle M_{p}^{2}\rangle$ of these fluctuations. (**d**) Without an applied electric field, orientations of the dipoles in a sample of polar fluid are random. The mean value, $\langle M_{T}\rangle {}_{0}$, of the total moment of these dipoles is thus zero. However, due to random fluctuations, its mean square value $\langle M_{T}^{2}\rangle {}_{0}$ is finite and corresponds to spontaneous polarization.

### 3.1. The Clausius–Mossotti Law, Lorentz Cavity Field and Debye’s Orientational Polarization

Based on Green’s electric potential function [7] and Mossotti’s hypothesis [9] that the electric fluid residing in each conductive ‘corpuscle’ of a dielectric is displaced under the action of a local field to form an electric doublet (i.e., dipole), Clausius derives the following relationship (known as the Clausius–Mossotti law) linking Faraday’s concept of a specific inductive capacity (i.e., the static dielectric constant) to a simplifying factor *g* that relates the induced moment of an individual corpuscle to the total polarization of the dielectric [10]:(24)$$\varepsilon_{m}=\left(1+2g\right)/\left(1-g\right)$$

The key steps to the derivation of this relationship are described elsewhere [24]. It is implicit that *g* is proportional to the packing number density, *N,* of corpuscles and is thus proportional to the mass density *ρ* of the dielectric. The following is an alternative and the usual way to express the Clausius–Mossotti law:(25)$$\left(\frac{\varepsilon_{m}-1}{\varepsilon_{m}+2}\right)\frac{1}{\rho }=\mathrm{constant}$$

This masks, but does not remove, the implied issue of the factor *g* approaching a value of unity, where either a small perturbation of temperature or applied field could result in infinite polarization and a ferroelectric transition. This is addressed in Section 3.3. Although Clausius postulates that each corpuscle could possess a permanent dipole moment, he describes their polarization as an induced elastic displacement of internal charges [10]. This is now known as atomic distortion polarizability, comprising the two distinct contributions of electronic and atomic polarizability. The number density *N* is equal to *N_A_ρ*/*M**_W_*, with *N_A_* the Avogadro constant, so that the polarization (induced moment per unit volume) P*_m_* is given in terms of the local field E*_L_* as:(26)$$P_{m}=N\alpha E_{L}$$

Lorentz evaluates E*_L_* by placing a polarizable particle inside a virtual spherical cavity “whose dimensions are infinitely small in a physical sense” [50]. All the atomic matter, apart from the point charges forming the dipole, are removed from this imagined cavity. The distributed charge induced at the cavity’s outer surface creates a cavity field E*_L_* of magnitude E*_m_* + P*_m_*/(3*ε*_o_). The atomic matter that had been removed is now returned, which adds another component *sP_m_*/*ε*_o_ to the cavity field, to give:(27)$$E_{L}=E_{m}+\left(\frac{1}{3}+s\right)\frac{P_{m}}{\varepsilon_{o}}$$

Lorentz shows, for a cubic lattice of polarizable particles, that *s* = 0, and states that otherwise this “is a constant that will be difficult exactly to determine” [50] (pp. 138, 303). This result also holds for an isotropic lattice, provided that no short-range interactions occur between the induced dipole fields. Substituting into Equation (27) the value for P*_m_* given by Equation (4) and assigning s = 0, E*_L_* has the value:(28)$$E_{L}=\frac{\varepsilon_{m}+2}{3}E_{m}$$

This is known as the Lorentz field (sometimes called the Clausius–Mossotti field). The susceptibility of the Lorentz cavity field, equal to the ratio E*_L_*/E*_m_*, is thus:(29)$$\chi_{Lc}=\frac{\varepsilon_{m}+2}{3}\;(s=0)\;\;\mathrm{or}\;\;\chi_{Lc}=\left[\frac{1}{3}\left(\varepsilon_{m}+2\right)+s\left(\varepsilon_{m}-1\right)\right]$$

Equation (29) indicates that the Lorentz cavity field, unlike the Maxwell cavity field of Equation (5), is always larger than the applied macroscopic field E*_m_*. This arises because the Lorentz cavity, being virtual and without a physical boundary between it and the surrounding dielectric, does not experience the field-shielding effect of induced interfacial charges as produced for the Maxwell cavity. From Equations (4), (26) and (28) the following relationship is obtained:(30)$$\frac{\varepsilon_{m}-1}{\varepsilon_{m}+2}=\frac{N\alpha }{3\varepsilon_{o}}$$

Lorentz recognizes that this relationship corresponds to the law formulated by Clausius as given by Equation (25). It can be used to derive good estimates of a dielectric’s refractive index *n* (*n*^2^ = *ε**_m_*). Debye extends its relevance to polar solutions, by adding the orientational polarization of a molecular dipole of moment *m* to the atomic distortion polarization [18]:(31)$$\frac{\varepsilon_{m}-1}{\varepsilon_{m}+2}=\frac{1}{3\varepsilon_{o}}\underset{i}{\sum }N\left(\alpha_{i}+\frac{m_{i}^{2}}{3kT}\right)=\frac{1}{3\varepsilon_{o}}\underset{i}{\sum }\frac{N_{A}\rho }{M_{w}}\left(\alpha_{i}+\frac{m_{i}^{2}}{3kT}\right)$$

The polar molecule is regarded as a sphere undergoing rotational Brownian motion, whilst experiencing the Lorentz cavity field inside a spherical cavity within the dielectric. The summation Σ*_i_* includes all types of polarizable molecule within the dielectric. In the absence of an applied electric field E*_m_*, the direction of orientations of an ensemble of dipoles will, on average, be distributed with the same probability over all directions in space. On applying a field, each dipole gains potential energy *U* = −*m*E*_m_* cos *θ*, where *θ* is the polar angle between the direction of the dipole and the applied field. Using classical Boltzmann statistical mechanics and a method resembling that of Langevin for gas molecules carrying a permanent magnetic moment [51], Debye obtains the following value for the average moment per dipole oriented along the applied field’s direction [18]:(32)〈m⟩E=m〈cos θ⟩=m23kTEm

Computer modelling of the dielectric properties of a solvated protein employs a lattice of such Langevin dipoles [52,53]. The value of 〈cos θ⟩ is given by the Langevin function, *L*(*x*) = coth x − 1/x, and is shown in Figure 9 [18,19].

In dielectric spectroscopy measurements the fields are usually less than 10^5^ V/m, so for many kinds of polar molecules (e.g., water) *L*(*x*) ≈ x/3 and values for 〈cos θ⟩ are appreciably smaller than unity (typically < 10^−4^). This indicates that very little change to the directions of the polarized dipoles has occurred compared with their random directions with no field applied. The magnitude of the orientational polarizability for each dipole is thus m23kT, and is incorporated as such in Equation (31). For protein iDEP experiments the applied fields are typically of the order ~10^6^ to 10^7^ V/m [4,5]. For many classes of polar molecule (e.g., water with *m* = 1.8 D), even with such high fields, the interaction energy, m*·*E*_m_*, is much less than the thermal energy *kT*. However, globular proteins possess dipole moments several hundred times larger than many other types of polar molecule [16,24]. As shown in Figure 9 for BSA (*m* = 710 D [54]) the linear relationship given by Equation (32) is no longer applicable and saturation of the orientational polarization is approached. With increasing field strength, 〈cos θ⟩ approaches unity, indicating that on average the protein’s permanent dipole moment is almost aligned with the field.

**Figure 9 micromachines-13-00261-f009:**
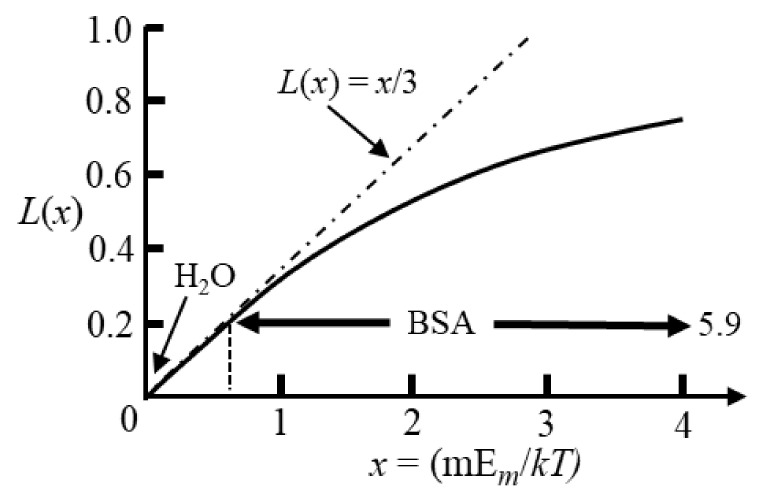
The average orientation of a single dipole along the direction of an applied field is given by the Langevin function (L(x) = coth x − 1/x) [18,19]. Fields of ~10^6^ to 10^7^ V/m are used in protein DEP experiments [4,5]. Polar molecules typically have dipole moments less than 5 debye units (e.g., 1.8 D for H_2_O) so that x << 1. Globular proteins have large dipole moments and as shown for BSA (m = 710 D [54]) saturation of its polarization can occur in DEP studies.

### 3.2. Maximum Size of a Lorentz Cavity

Ramshaw offers a basis for defining the maximum size permitted for a virtual Lorentz cavity [55,56]. He makes the interesting observation that in Equation (4) both P*_m_* (dipole moment per unit volume) and E*_m_* (the macroscopic Maxwell field) can be defined at any location within a dielectric, regardless of whether *ε_m_* exists as a real concept. Thus, the existence of *ε_m_* cannot be guaranteed, by definition. What does this mean within the context of the Lorentz cavity? Ramshaw [55] addresses this by expanding the Clausius–Mossotti function of Equation (25) into powers of the number density, *N,* of polar molecules, keeping the material’s specific density *ρ* constant. This is equivalent to finding the virial coefficients for a gas [18]. Ramshaw makes no assumption regarding *ɛ_m_* and also eliminates *E_m_* from his analysis by considering both short-range and long-range orientation correlations of pairs of dipoles. The total correlation between two dipoles, in zero applied field, is regarded as the sum of a direct and an indirect effect that is transmitted by chains of direct effects between intermediate molecules. Beyond a certain total chain length *d*, where dipole–dipole intermolecular potential energies fall below *kT*, thermal disruption overcomes coherent correlations of dipole pairs. If a cavity’s diameter is smaller than *d*, *ɛ_m_* can be considered as well-defined and the cavity field is the Lorentz field, E*_L_*, given by Equation (27). Above this cavity size, no such assumption may be made [55,56].

### 3.3. Onsager’s Reaction Field [57] and Kirkwood’s Correlation Factor [58,59]

Onsager, a PhD student of Debye, decided to learn ‘about the dielectrics that Debye had done’. He was ‘too lazy to go to the libraries and sat down and worked it out and lo and behold it came out quite different!’ [60]. Onsager replaces the cavity used by Lorentz and Debye with a much smaller one, of the same radius, R, as the polar molecule of interest. The effect is considered of introducing a rigid point dipole of moment m_o_ into this cavity, with no external field applied. The thermal average <m_o_> of this dipole moment is calculated to seamlessly match it to the known number density of dipoles in the surrounding medium. The point dipole polarises the surrounding homogenous medium, creating interfacial charges and a reaction field inside the cavity. Matyushov considers the physics of this as two charged lobes, at the opposite sides of the void containing the point-dipole, visualized as oriented dipoles of the medium cut through by the dividing surface [28]. Adopting Matyushov’s nomenclature, the interface dipole created by the reaction field is:(33)Mint=−2εm−12εm+1mo

The minus sign signifies that the reaction field acts against the field created by the fixed dipole moment, m*_o_*, and almost cancels it out for polar media of high permittivity [28]. Onsager’s introduction of the reaction field removes the potential problem of the factor *g* approaching a value of unity in Equation (19).

The Kirkwood correlation factor, *g_k_*, quantifies the deviation from randomness of the orientation of a dipole, with respect to its neighbours, in a liquid composed of polar molecules. If *N* is the number of dipoles, each of moment *m*, in an ensemble of dipoles, then *g_k_* is given by:(34)gk=〈M2〉Nm2
where $\langle M^{2}\rangle$ is the mean square value of the ensemble’s total dipole moment, M. The significance of *g_k_* is demonstrated in Figure 4, where it addresses the fact that Equation (31) incorrectly predicts water at room temperature to possess a static permittivity, *ε_m_* ≈ 31, instead of the experimental value, *ε_m_* ≈ 78. In its condensed phase, moment *m* (corresponding to that of an isolated H_2_O molecule) is enhanced due to its orientational fluctuations being amplified through coordinated rotations of dipole neighbours to which it is hydrogen bonded. Kirkwood replaces Σ*m*^2^ in Equation (31) with the summation Σ<m*_o_*·m> (i.e., Σ m*_o_*m cos *θ*) with *θ* being the angle between a target dipole, m*_o_*, and a neighbouring dipole, m. The correlation factor *g_k_* is defined as the average of cos *θ* between this target dipole and all the dipoles (including itself) within a volume of polar liquid surrounding it [58,59]. The summation thus includes <m*_o_*·m*_o_*>, which is given the value of unity (*θ* is effectively zero, so cos *θ* = 1). If there is no correlation at all between the dipoles, then <cos *θ*> = 0 and *g_k_* = 1.0. Hydrogen bond links between neighbouring dipoles in bulk water leads to correlation. For a perfect and rigid tetrahedral coordination of the H-bonds, about which free rotations are permitted but no bond bending, *g_k_* is given as
(35)$$g_{k}=1+z\cos^{2}\frac{\theta }{2}$$
where *θ* is the H-O-H bond angle and *z* is the coordination number. Based on coordination numbers (*z* = 4 for a perfect tetrahedron) and inter-molecular distances obtained from X-ray scattering data for water, the value *g_k_* = 2.67 was obtained, to give *ε_m_* = 78.2 [61].

Based on these contributions by Onsager [57] and Kirkwood [58,59], Equation (31) evolves to the general form [16]:(36)$$\frac{\left(\varepsilon_{m}-\varepsilon_{\infty }\right)}{\varepsilon_{m}}\frac{\left(2\varepsilon_{m}+\varepsilon_{\infty }\right)}{{\left(\varepsilon_{\infty }+2\right)}^{2}}=\frac{1}{3\varepsilon_{o}}\underset{i}{\sum }N\left(\alpha_{i}+\frac{g_{k}m_{i}^{2}}{3kT}\right)$$
where *ɛ_m_* is the static or ‘equilibrium’ permittivity value (corresponding to where the system of dipoles has attained a constant value of polarization with a constant applied field), and $\varepsilon_{\infty }$ is the square of the refractive index at high frequencies ~10^13^ Hz [20]. Typically, for pure polar liquids, $\varepsilon_{m}\gg \varepsilon_{\infty }$ with $\varepsilon_{\infty }\approx 1$, so that for cases where only the dipole polarization is of interest (i.e., the atomic distortion polarizability *α_i_* is not included) Equation (36) is usually presented as
(37)$$\frac{\left(\varepsilon_{m}-1\right)\left(2\varepsilon_{m}+1\right)}{9\varepsilon_{m}}=N_{dip}g_{k}$$
where *N**_dip_* is the dipole polarizability density given by
(38)Ndip=Nm29ε0kT
with *N* being the number of polar molecules per unit volume. Equation (37) applies to the static case. If we wish to consider an orientational dipole dispersion, Δ*ε*, occurring between a low and high frequency limit, the static permittivity, *ɛ_m_*, is replaced by the complex permittivity, $\varepsilon \left(\omega \right)=\varepsilon^{\prime }-i\varepsilon^{\prime \prime }$, of the polar liquid at frequency ω/2π. On applying a sinusoidal field, E(*ω*) = E*_pk_* exp(*iωt*), to the system of dipoles, the variation of the permittivity as a function of frequency is given through the Laplace transform [62]:(39)$$\frac{\left[\varepsilon \left(\omega \right)-1\right]\left[2\varepsilon \left(\omega \right)+1\right]}{9\varepsilon \left(\omega \right)}=\left\{\frac{\left[\varepsilon_{m}-1\right]\left[2\varepsilon_{m}+1\right]}{9\varepsilon_{m}}\right\}\;L\left\{-\varphi_{M}\left(t\right)\right\}$$

The Laplace transform is defined as [63]:(40)$$L\left\{f\left(t\right)\right\}=\int_{0}^{\infty }f\left(t\right)\exp\left(-i\omega \tau \right)dt$$
and in this case $-\varphi_{M}\left(t\right)$ is the autocorrelation (decay) function of the permanent dipoles. This represents a combination of the superposition theorem with Onsager’s theory that microscopic fluctuations created by thermal energy obey the same law of decay as macroscopic perturbations produced by an applied field [62]. According to Onsager’s proposal [64] the following auto-correlation functions are equivalent:(41)$$\varphi_{M}\left(t\right)=\frac{{\langle M\left(t\right)\rangle }_{o}}{{\langle M\left(0\right)\rangle }_{o}}\;(\mathrm{Decay}\;\mathrm{function}\;\mathrm{of}\;a\;\mathrm{macroscopic}\;\mathrm{moment});\;\varphi_{M}\left(t\right)=\frac{{\langle M\left(t\right)\cdot M\left(0\right)\rangle }_{0}}{{\langle M\left(0\right)\cdot M\left(0\right)\rangle }_{0}}\;(\mathrm{Correlation}\;\mathrm{function}\;\mathrm{of}\;\mathrm{moment}\;\mathrm{fluctuations})$$

If these are expressed through a sum of exponential terms ∑i=1naiexp−tτi, the Laplace transform is the sum of equivalent Debye terms:(42)L−φwt=∑i=1nai1+iωτi
where *a**_i_* is the amplitude, Δ*ε*(*i*), and *τ* is the characteristic dipole correlation time for each dispersion. Δ*ε*(*i*) is the difference of real components of *ε*(*ω*), so that each dispersion is characterized by the frequency variation of *N**_dip_* of Equation (38), namely:(43)Ndipω=Ndip1+ω2τi2

This is the origin of the frequency dependence given for *ε_m_*(*f*) in Equation (17).

### 3.4. Fröhlich’s Theory Relating Permittivity and Spontaneous Polarization

This theory is used in molecular dynamics (MD) simulations of solvated proteins. Fröhlich [19] (p. 37) considers a macroscopic spherical region of volume *V*, within an infinite homogeneous dielectric medium. Volume *V* is large enough to exhibit the same static permittivity, *ε_m_*, as the surrounding homogeneous medium (refer to discussion after Equation (12)). The volume’s surface can deviate, so that no molecule is cut by the surface, and encloses an ensemble of dipoles. Each dipole is represented as an elementary charge that can be thermally activated to ‘hop’ between two adjacent potential energy wells (see [24] for a schematic). The position of each charge is described as the vectorial displacement from the position it would have in the lowest energy level (ground state) of the whole system, and given according to the rules of Boltzmann statistical mechanics. However, the medium outside the sphere is treated as a continuous dielectric described by its macroscopic permittivity. Except at the absolute zero of temperature, the ensemble of charges inside the sphere will not retain a fixed positional configuration—even if macroscopically it is in equilibrium. Owing to thermal fluctuations there is a probability of finding this ensemble with any set of displaced space elements. This means, despite the average moment, ${\langle M\rangle }_{0}$, of the ensemble in the absence of a field being zero, its mean square value, $\langle M^{2}\rangle {}_{0}$, in the absence of a field, has a finite value (see Figure 8d).

Fröhlich adopts the Onsager reaction field as the ‘self-field’ of the spherical volume and then adds to this the Maxwell cavity field produced by an external field E. The external field is considered to be sufficiently weak that the non-linear (saturation) polarization region shown in Figure 9 is avoided, so that $\left(M\cdot E\right)/kT\ll 1.$ Furthermore, Onsager’s ‘self-field’ is shown by Equation (33) to be the dominant component of the local field. Fröhlich’s conclusion that the static permittivity of a sphere of a dielectric embedded in a large specimen of the same material can be expressed in terms of its spontaneous polarization, can thus be anticipated. He derives the following expression for the dielectric’s susceptibility:(44)$$\left(\varepsilon_{m}-1\right)=\frac{\langle M^{2}\rangle {}_{o}}{3kT\varepsilon_{o}V}\left[\frac{3\varepsilon_{m}}{\left(2\varepsilon_{m}+1\right)}\right]=\frac{\langle M^{2}\rangle {}_{o}}{3kT\varepsilon_{o}V}\chi_{Mc}$$
where $\langle M^{2}\rangle {}_{o}$ indicates the Boltzmann average in the absence of a field, and from Equation (6) the susceptibility, *χ_Mc_*, of the Maxwell cavity field is identified.

Fröhlich generalizes his model by permitting the immersed spherical volume to have an arbitrary static permittivity *ε*_p_. For this ‘mixture’ model, he obtains the following result [19] (p. 177):(45)$$\left(\varepsilon_{p}-1\right)=\frac{\langle M_{p}^{2}\rangle {}_{o}}{3kT\varepsilon_{o}V_{p}}\left[\frac{\left(2\varepsilon_{m}+\varepsilon_{p}\right)}{\left(2\varepsilon_{m}+1\right)}\right]=\frac{\langle M_{p}^{2}\rangle {}_{o}}{3kT\varepsilon_{o}V_{p}}\chi_{Mcp}$$
where the Maxwell cavity susceptibility, *χ_Mcp_*, is given by Equation (5). The fact $\langle M_{p}^{2}\rangle {}_{o}$ occurs in the absence of an applied field indicates that the macroscopic sample of volume *V**_p_* can be of arbitrary shape. Matyushov expresses this as an ‘invariant of rotations of the laboratory frame’ [28]. Simonson adopts Equation (45) in a molecular dynamics (MD) simulation at 295 K of a solvated cytochrome-c molecule contained within a 2.4 nm radius sphere of water [31]. His conclusion that *ε_p_* ≈ 25 was used to give the $\nabla E_{Th}^{2}$ values shown in Figure 2.

### 3.5. South and Grant’s Theory of Protein Dipole Relaxation

South and Grant [65] apply Fröhlich’s model to a sphere of permittivity $\varepsilon_{\infty w}$ and volume *V* containing *N_w_* water molecules and *N_m_* protein molecules. Each molecule is represented as a point dipole, *m_w_* for the water and *m_p_* for the protein, suspended in a ‘background’ medium of effective permittivity, $\varepsilon_{\infty w}$. This ‘background’ is responsible for spontaneous fluctuations of induced moments. The sphere is immersed in an infinite continuum of permittivity *ε_mix_*, i.e., the same as the effective *static* permittivity of the sphere itself. The spontaneous fluctuations of induced and permanent dipoles are assumed to arise from independent mechanisms [66]. The short-range protein–water and water–water correlations are represented by Kirkwood factors *g_kp_* and g*_kw_*, respectively, and are assumed to be insensitive to protein concentration. Because each molecule is represented as a point dipole, no account is taken of the fact that a globular protein molecule is several thousand-times larger than the water molecules. South and Grant take this into account by representing the water as a continuum, thus removing the water dipoles from their model but retaining their influence by replacing the background medium by one whose permittivity is the static value permittivity, *ε_sw_*, of pure water.

The following relationship is derived for the model of a central sphere of effective permittivity $\varepsilon_{\infty w}$ and volume *V* containing *N_p_* protein molecules, each of effective dipole moment *m_p_* and surrounded by a hydration shell of permittivity *ε_mw_*, in an infinite continuum of static permittivity *ε_mix_*:(46)$$\left(\varepsilon_{mix}-\varepsilon_{sw}\right)=\frac{N_{p}g_{kp}m_{p}^{2}}{3kT\varepsilon_{o}V}\left[\frac{3\varepsilon_{mix}}{\left(2\varepsilon_{mix}+\varepsilon_{sw}\right)}\right]$$

It is assumed that *g_kp_* ≈ 1 for the situation where the overall short-range interactions of a protein with water molecules are nearly spherically symmetrical.

Following the procedure leading to Equation (39), the magnitude of the *β*-dispersion exhibited by a dilute protein solution is [65]:(47)$$\Delta \varepsilon \left(\beta \right)=\frac{hN_{A}C_{p}}{\varepsilon_{0}M_{w}kT}g_{kp}m_{p}^{2}$$
where the number density *N**_p_*/*V* has been replaced by *N_A_C**_p_*/*M**_W_*, with *N_A_* the Avogadro constant, *M**_W_* the protein’s molecular weight and *C**_p_* its concentration (g/mL). The value of *h* is ½ when the solvent is considered to be an assembly of point dipoles, and *h* = 3/4 when the continuum approximation is made [65]. This result can be compared with the following semiempirical relationship formulated by Oncley [67]
(48)$$\Delta \varepsilon \left(\beta \right)=\frac{bN_{A}C_{p}}{9\varepsilon_{o}M_{w}kT}m_{p}^{2}$$

The parameter *b* is calibrated by Oncley to be 5.8, based on dipole moment values obtained for simple amino acids by determination of the separation distance of the positive and negative charges carried by their amino and carboxyl groups. However, Takashima and Asami [68] found that a value for *b* = 4.5 gives better agreement between the calculated and experimental dipole moment for a solvated protein, if a linear relationship between Δ*ɛ*(*β*) and *C_p_* is found. Based on *b* = 4.5, from Equations (47) and (48) estimates for *g**_kp_* of 1.0 (*h* = 1/2) and 0.67 (*h* = ¾) arise.

Values for *g_kp_* less than unity imply anticorrelation of the protein and water dipoles of hydration, and from MD studies this was concluded to be the case for the inner population of water dipoles around ubiquitin and apo-calbindin [69,70]. The inner population of water, less than 0.35 nm from the protein surface, consists of water patches around polar and charged amino side-groups, whereas almost no waters are found this close to the hydrophobic groups [69]. Water molecules in the outer hydration shell (0.35~0.6 nm from the surface) either form hydrogen bonds with the inner bound waters or cover the hydration ‘holes’ about the non-polar and uncharged areas of the protein surface. In the MD simulations, ubiquitin was found to slow down the dynamics of bulk water molecules located as far away as 1.35 nm from its surface [69].

## 4. Heyden and Matyushov’s Theory and MD Simulations

### 4.1. The Theory

This builds on foundations laid down by Matyushov and co-workers [71,72,73,74]. A key aspect is to replace the dielectric boundary-value problem of macroscopic electrostatics by calculation of the cross correlation of the protein’s permanent dipole moment with its polarized hydration shell. The result is replacement of *Re*[*CM*]*_macro_* with the factor *K,* so that Equation (10) now takes the form:(49)$$\chi_{DEP}=\frac{3}{2}\;V_{p}\varepsilon_{m}\;Re\left[K\right]$$

As depicted in Figure 1, *K* holds the key to making the transition from the standard DEP theory to the new one, (from [*CM*]*_macroscopic_* to [*CM*]*_molecular_*), and is given by:(50)$$K=\varepsilon_{m}\chi_{c}^{HM}\frac{N_{dip}}{3}+\frac{3\varepsilon_{m}}{2\left(\varepsilon_{m}-1\right)}\left(\chi_{c}^{HM}-\chi_{Lc}\right)$$

The susceptibility factor $\chi_{c}^{HM}$ is new to dielectric theory (where the superscript ‘*HM*’ signifies its genesis) and effectively replaces Oncley’s empirical parameter, *b*, of Equation (48). *χ_Lc_* is the Lorentz cavity field susceptibility of Equation (6) and *N_dip_* is the static dimensionless number density of Equation (36). For a particle without a permanent dipole moment, then *N_dip_* = 0 and *K* is equivalent to [*CM*]*_macroscopic_*. For the case where *N_dip_* is finite and *K* effectively acts as [*CM*]*_molecular_*, calculation of $\chi_{c}^{HM}$ is required and is given by [1]:(51)$$\chi_{c}^{HM}=\chi_{c}^{HM}\left(\alpha \right)\left(\langle M_{p}\cdot M_{t}\rangle /\langle M_{p}^{2}\rangle \right)=\left\{\left(\langle M_{p}^{2}+\langle M_{p}\cdot M_{w}\rangle \rangle \right)/\langle M_{p}^{2}\rangle \right\}$$

This calculation, performed in the MD simulation, derives the direct correlation between the dipole moment of the protein and its hydration shell, M_p_, and the dipole moment, M_w_, induced in the surrounding aqueous medium. This contains the self-variance, $\langle M_{p}^{2}\rangle$, and the cross-correlation, $\langle M_{p}\cdot M_{w}\rangle ,$ between the hydrated protein and bulk water dipoles. $\chi_{c}^{HM}$ is given by
(52)χcHMα=χLc−α2εm−123εm2εm+1

The factor *α* can be assigned two values, namely zero or unity. For *α* = 0, $\chi_{c}^{HM}$ is equal to the Lorentz cavity susceptibility, *χ_Lc_*. The surface between an inner and outer volume of dielectric does not produce the interfacial dipole moment M^*int*^ of Equation (4). The assignment of *α* = 1 brings with it a physical interface and creation of M^*int*^, with Equation (52) now given by:(53)χcHMα=1=32εm+1

In this case, $\chi_{c}^{HM}$ represents the susceptibility of the Maxwell cavity. For reasons clearly explained [1], in deriving Equation (53) from Equation (5), the Maxwell field E*_m_* is E*_0_*/*ε_m_*, where E*_0_* is the vacuum field. As indicated in Equation (51), calculation of not only the self-variance, $\langle M_{p}^{2}\rangle$, of the protein’s moment is involved, but also that of the cross correlations between M_p_ and the total moment, M_t_, of the water dipoles comprising the surrounding dielectric continuum. Of particular importance is the cross correlation involving the moment, M^*int*^, associated with polarization of its hydration shell [1]. The standard macroscopic boundary conditions of macroscopic electrostatics, used to derive the DEP susceptibility values given by Equations (6) and (10), apply only for the case where the influence of a particle’s induced moment dominates over that of its permanent dipole moment (should it possess one).

The following equation is derived for the collective dielectric dispersions of a protein solution:(54)$$\Delta \varepsilon_{mix}\left(\omega \right)=\left(\frac{9}{2}\right)\nu_{p}N_{dip}\left(\omega \right)+9\nu_{p}N_{dip}\left(\omega \right)\left[\chi_{c}^{HM}\left(\omega \right)-1\right]+\left(1-\nu_{p}\right)\Delta \varepsilon_{w}\left(\omega \right)$$
with *v_p_* the volume fraction of the protein and *N_dip_*(ω) given by Equation (43).

The first term in Equation (54) is the *β*-dispersion, Δ*ε*(*β*), produced by protein tumbling, followed next by the *δ*-dispersion, Δ*ε*(*δ*), that is assumed to be associated with the protein hydration sheath (see Section 2.2). The magnitude and frequency dependence of Δ*ε*(*β*) is in line with standard dielectric theory, but the expression given for Δ*ε*(*δ*) is new. The last term in Equation (54) is the *γ*-dispersion for the relaxation of bulk water dipoles and is attenuated by the factor (1-*v_p_*) representing the partial volume of low polarizability occupied by the non-relaxing proteins and their hydration shells. It is interesting to note that if $\chi_{c}^{HM}=1$, corresponding to no correlations between the protein dipole and water dipoles in its hydration shell, then Δ*ε*(*δ*) vanishes. Written in terms of dipole moments of the protein and water, the Δ*ε*(*δ*) term in Equation (54) becomes
(55)9νpNdipχcHM−1=νpMp·MwVpε0kT

The dispersion strengths Δ*ε*(*β*) and Δ*ε*(*δ*) are given by
(56)$$\Delta \varepsilon \left(\beta \right)=\frac{9}{2}\nu_{p}N_{dip}$$
(57)$$\Delta \varepsilon \left(\delta \right)=9\nu_{p}N_{dip}\left(\chi_{c}^{HM}-1\right)$$
to give
(58)χcHM=1+Δεδ2Δεβ

Based on experimental values reported for the magnitudes of the Δ*ε*(*β*) and Δ*ε*(*δ*) dispersions [4,24,44,68], as well as inspection of Figure 5, it is apparent that the term in brackets is small, indicating that $\chi_{c}^{HM}\approx 1$. In other words, the protein–water Kirkwood correlation factor is close to unity, implying very little correlation, in line with South and Grant’s assumption [65].

The DEP response of a solvated protein as a function of frequency, as depicted in Figure 5 for BSA, is primarily related to Δ*ε*(*β*) and thus through Equations (36) and (56) depends on *N_dip_*(*ω*). The following relationship to connect *K* and Δ*ε*(*ω*) is derived [1]:(59)Kω=29εωΔεωνp+ΔεwωχcHMω2χcHMω−1

From the MD simulations, the value of the term in square brackets is estimated to be close to unity. For dilute protein concentrations, as *v_p_* tends to zero, the term Δε(ω) can also be neglected, so that to a good approximation:(60)Kω=29εωΔεωνp

The DEP frequency response is given by Re[*K*] in Equation (49) and is shown in Figure 10 for lysozyme and ubiquitin as calculated from Equation (60) and based on the MD simulations [1]. Included in this figure are the empirical factors $\left(\kappa +2\right){\left[CM\right]}^{empirical}$ derived elsewhere for these two proteins [4] (see Figure 6 for estimated DEP cross-over frequencies). When normalized, the calculated and empirical results for ubiquitin are similar, reflecting a close correspondence of the MD simulation [1] and the experimental spectroscopy data [75] used to derive $\left(\kappa +2\right){\left[CM\right]}^{empirical}$ [4]. The difference shown for the high-frequency tail of lysozyme results from a disparity of the experimental data [44] and the MD simulation—indicating the importance of the latter to molecular DEP studies. The difference in magnitudes shown for ubiquitin is related to the new theory [1] replacing Oncley’s empirical factor (*b* = 4.5) of Equation (48) with the susceptibility factor $\chi_{c}^{HM}$ of Equation (51), together with the fact that the empirical theory [4] employs mass density values for the protein [4] and experimental values for Δ*ε*(*β*)/*v**_p_*.

### 4.2. Molecular Dynamics Studies

In brief, the goal of the MD simulations of ubiquitin and lysozyme is to determine the cavity susceptibility given by Equation (51), relating the combined dipole moment, M_p_, of the protein and its hydration shell to the dipole moment induced in the surrounding solution. When this equation is applied to the simulation trajectories, M_p_ is calculated within a sphere of cut-off radius, r_c_, drawn around the protein. A correction factor, previously found to be independent of r_c_ [71], is applied to account for the fact that at each configuration along the simulation trajectory the outside solution is polarized by the dipole M_p_.

Two features of these MD simulations are noteworthy and represent clear advances, namely their timescales and the atomic contents of the simulation cells. Simulations with integration steps of either 1 or 2 femtoseconds were performed for run times of 1 and 10 microseconds. In the frequency domain, a simulation run of 10 μs at steps of 1 fs corresponds to the range from 10^5^ to 10^12^ radians/sec (i.e., 16 kHz to 160 GHz). This permits analyses to be made of the β, *δ* and *γ* relaxations exhibited by medium-sized proteins of molecular weight up to 100 kD (see Figure 4). By comparison, with the computer power available to them at the time, Boresch et al. [76] were limited to simulation runs of 5 ns at 2 fs steps in their pioneering MD simulation of ubiquitin. With such run times, adequate characterization of the *δ*-dispersion exhibited by globular proteins is possible, but only for proteins of low molecular weight (e.g., ubiquitin, cytochrome-c, lysozyme and myoglobin) is partial investigation of the β-dispersion achievable. Even so, these simulations were able to clarify which mechanisms (self- or cross-correlation) are responsible for the β- and *δ*-relaxations [76,77,78]. In subsequent MD simulations extending to 15 ns [69,70], ubiquitin, apo-calbindin D-9K, and the C-terminal SH2 domain of phospholipase C were studied to represent the structural variability found in medium-sized proteins. Two hydration shells were clearly discerned about charged and polar amino acids, and it was concluded that the protein molecule slows down the dynamics of water molecules located as far away as 1.35 nm [70]. The extension from MD run times of 15 nanoseconds to 10 microseconds [1] represents a significant advance. It also permits a significant increase in simulation cell size and content to be made.

The simulation cell (a cube of sides 12.5 nm) contained atomic copies of the proteins, based on high resolution x-ray crystal structures, together with 64,139 and 64,155 water molecules for ubiquitin and lysozyme, respectively, that included those resolved in their crystal structures [1]. The ionization (protonation) states of the titratable amino acid sidechains were determined for pH 7. The resulting total charge of ubiquitin was zero, and the charge (+8) of lysozyme was neutralized by a uniform counter charge. As a starting point for a 1 microsecond MD simulation, the energy of each system was minimized to a local minimum to avoid clashes between the protein and added water molecules, and then equilibrated at 300 K and 1 bar for 0.2 microseconds. These MD simulations require massive computing power. As an estimate, a simulation run of 1 microsecond at steps of 1 femtosecond, where the movements of around 200,000 atoms are determined, requires in excess of 10^14^ calculations to be made!

## 5. Something Else?

DEP experimentation and protein sample preparation fall under this heading. The published protein DEP data has mainly been obtained using either microfabricated metal electrodes to create the field gradients, referred to as eDEP, or iDEP where insulating microstructures are employed for this purpose. The eDEP experiments have reported results in line with those expected of the empirical [4] and formal theory [1]. However, inconsistent results (even for the same protein) have occurred for the iDEP experiments, and it is here where interesting and potentially exploitable examples of ‘Something Else’ exist. Some of these inconsistencies may be associated with iDEP departing from experimental guidelines commonly adopted for eDEP.

### 5.1. eDEP

The first eDEP experiments (and use of the term ‘dielectrophoresis’) were performed on macroscopic particles and employed metal electrodes in the form of tungsten wire and tinfoil, for example [13]. In early eDEP studies using *microfabricated* metal electrodes to study bacteria and cells [79,80,81,82,83] the following electrokinetic and related phenomena were observed and reported:Below ~1 kHz the DEP response of bacteria is dominated by a surface conductivity related to mobile ions in their surface electrical double layer [79,80].Electrothermal fluid motion can disrupt DEP at frequencies below ~10 KHz [80].Below ~1 kHz electrode polarization should be taken into account [81].Below ~100 Hz the electrophoresis and DEP of bacteria are superimposed and enhance positive DEP [81].Below ~100 Hz the DEP of mammalian cells rises rapidly with decreasing frequency. Neuraminidase treatment confirms this is associated with cell surface charge [82].Below ~500 Hz particles undergoing negative DEP are driven onto the surface of planar electrodes [83]. This is later shown to be caused by AC electroosmosis [84,85].

It is instructive to examine the procedures adopted for the first report of protein DEP by Washizu et al. [2]. They observed positive DEP of fluorescently labelled proteins (avidin, chymotripsinogen, concanavalin and ribonuclease) using microelectrodes fabricated from either aluminium or platinum. Measurements in the frequency range 1 kHz–10 MHz were performed, with the proteins suspended in their dialysis media of conductivity around 0.3 mS/m. Gel chromatography was used to verify that the samples after dialysis were in monomer form, and the similarity of results obtained using aluminium or the less electrochemically active platinum electrodes indicated that generated multivalent positive ions had not enhanced protein aggregation. The fact that DEP of the proteins was observed at a threshold field factor, $\nabla E_{Th}^{2}$, much lower than that indicated in Figure 2 could possibly have resulted from pearl-chaining of the proteins. Such chaining arises from dipole–dipole interactions, which should be proportional to d^−4^ (where d is the initial spacing between dipoles). This was eliminated as a contributing factor by the similarity of the dynamics of the DEP responses for initial protein concentrations of 0.01 μg/mL and 0.1 μg/mL. However, possible evidence that protein aggregation occurred at the higher initial concentration of 1 μg/mL and for frequencies around 1 kHz was indicated by a ‘memory’ phenomenon. Samples that had already experienced DEP either exhibited a lower threshold field value for the next measurement, or the DEP force acting on them gradually increased with time of exposure to a constant field. Fluid motion arising from Joule heating was also investigated and eliminated as a possible artifact. In their concluding comments, Washizu et al. [7] state:
*“although we may be seeing the combination of agglomeration and DEP at the low-frequency region, agglomeration is not a prerequisite for molecular accumulation, and DEP does occur with protein monomers.”*


Based on this careful work of Washizu et al. [2] the following can be proposed as guidelines for observing *positive* DEP of protein *monomers*:(i)Validate (e.g., through gel chromatography) that the samples are protein monomers.(iiLimit protein concentrations to below ~0.1 μg/mL (i.e., less than 7 μM for many candidate proteins).(iii)Adopt 1 mS/m as the upper limit for the aqueous solvent conductivity.

In protein eDEP experiments these guidelines have largely been followed. Solution conductivities of 1 mS/m or lower have been employed, with protein sample concentrations mostly below 10 μM. For all proteins investigated a consistent observation of *positive* DEP has been reported for the frequency range 1 kHz–1 MHz, in line with expectations of the empirical [4] and formal [1] theories. For some proteins a cross-over to negative DEP has been reported in the range 1–10 MHz [4], a finding consistent with the proposal that the DEP response for proteins can be predicted in terms of its dielectric β-dispersion [3]. However, the reported protein iDEP data for proteins is not consistent, and so it is here that categories falling under the umbrella of ‘Something Else?’ can be suggested.

### 5.2. iDEP

As introduced by Cummings and Singh [86] and reviewed by its practitioners [87,88,89] the complications that electrophoresis and electroosmosis can bring to eDEP experiments, especially at frequencies below 1 kHz, are translated through iDEP into facilitators of novel microfluidic devices for the selective sorting or concentration of molecular and macroscopic bioparticles. Negative DEP manifests itself as “streaming DEP” where the particles are carried down an array of insulating posts, following stream lines dictated by the spacing and geometry of the posts and the induced electroosmotic fluid flow. Streaming DEP occurs when the DEP force overcomes particle diffusion but is weaker than the combined effects of electrophoresis and electroosmosis. Increasing the magnitude of the externally applied field can change streaming DEP to “trapping DEP”, where the DEP force dominates over diffusion as well as electrokinetic flow and target particles can be reversibly immobilized at the insulating posts [86]. This introduces increased difficulties of interpreting particle motions, especially for submicron particles, where a host of potential linear and non-linear electrokinetic and electrothermal effects are available for consideration [90].

Examples can be cited where iDEP of macroscopic particles has been successfully interpreted in terms of the standard DEP theory. Kim et al. [91], in a review of DEP studies of bioparticles below the size of a typical mammalian cell (e.g., microbes, organelles, exosomes, nucleic acids and proteins), provide a comprehensive summary of the ‘one core’, single- and multi-shell models that have been applied to spherical and non-spherical bioparticles. While microbes can in many cases be described using such classical DEP models, Kim et al. conclude that the theoretical basis of protein- and DNA-iDEP requires further fundamental studies to allow the prediction of biomolecular DEP response and hence their tailored applications. Using DC-iDEP, Lapizco-Encinas et al. [92] separated live from dead *E. coli* and isolated Gram-positive from Gram-negative bacteria. However, the spatial separation of different bacteria resulted from differences of their exhibited negative DEP mobility—the separation by both positive and negative DEP previously reported using eDEP (10–100 kHz) [93,94] to physically separate target bacteria from a mixture was not possible. This result could be consistent with a DEP force determined by polarizations of a bacteria’s electrical double-layer, where the Dukhin function now serves as the effective [*CM*] factor [24]. In another study, by controlling the electroosmotic flow through a central outlet port and side outlet channels in a DC-iDEP device, the negative DEP force acting on the larger crystals in a mixture of protein crystals (size range ~80 nm to 20 μm) was sufficient to direct them into a central fluid outlet [95]. The weaker DEP force acting on the smaller crystals (size range ~80–200 nm) resulted in them remaining in the main stream emerging from the side outlets. In another example of controlled iDEP for macroscopic particles, polystyrene beads and yeast cells were focused into a thin stream line using an applied field in the form of a 0.1 Hz square wave. In this situation the negative DEP force remained unchanged in direction but there was a periodic reversal of the electroosmotic and electrophoretic forces [96].

Inconsistent iDEP results for proteins occur when the field is generated using a direct current (DC) voltage. Voltages as high as 4000 V have been used to concentrate PEGylated RNase by positive DEP [97,98]. However, the DEP force acting on the native protein was considered to be too weak to overcome electroosmosis [97]. An interesting situation is shown in Figure 11a where BSA samples, at the same pH and buffer conductivity, exhibit either negative [21] or positive [26,99] DEP. The concentration of the sample exhibiting negative DEP sample is 0.15 mM—much higher than the guideline of less than 10 μM adopted for eDEP studies. Positive DEP was observed for a BSA concentration of 7 nM [26,99]. The tendency of a solvated protein to dimerize or aggregate to form small crystallites is concentration-, temperature-, pH- and buffer-sensitive [4,100,101]. For example, BSA can be driven to a metastable state at a high mass concentration and high values of buffer pH and conductivity. In this respect, it is also of interest to note that in their iDEP experiments with high protein concentrations, Liu and Hayes [23] reported negative DEP responses for α-chymotrypsinogen (4 mM), immunoglobulin G (0.7 mM) and lysozyme (7 mM). Proteins in an iDEP experiment can also be subjected to shear stresses and high electric field strengths, factors known to influence initial protein crystal growth [102,103]. A field driven mechanism (possibly involving pearl chaining) whereby proteins in a metastable state may crystalize is proposed in Figure 11b. As shown in Figure 2, crystallites of diameter 50~100 μm are large enough to exhibit DEP according to standard theory. This is an aspect of protein DEP within the mesoscale between [*CM*]*_macroscopic_* and [*CM*]*_molecular_*, where *κ* or *N_dip_*, respectively, may or may not equal unity in either Equation (21) or (50), respectively.

As reviewed by Hill and Lapizco-Encinas [104], significant efforts have been made to mathematically model iDEP-based microfluidic devices and to identify empirical correction factors that can be added to align model predictions with experimental observations. These correction factors are intended to take account of electrothermally induced fluid flow, Joule heating, particle–particle interactions and temperature gradients, for example. For micron-sized particles (e.g., bacteria, blood cells, polystyrene beads and yeast cells) most correction factors are found to be small (0.3 to ~15), whilst particles of diameter ~1 μm attract larger correction factors—in some cases as large as 500~600, depending on the geometry and layout of the insulating posts, hurdles or restrictions [104].

The electrokinetic forces of electrophoresis and electroosmosis that, together with DEP, act on a particle in an iDEP experiment are traditionally assumed to have a linear dependence on the applied electric field. Their superposition is commonly referred to as ‘linear electrokinetics’. However, as recently reviewed [89,91] an increased understanding of how insulators alter the magnitude and non-uniformity of an applied electric field has led to a realization of the importance of non-linear electrokinetic effects [105,106,107,108,109]. For example, a nonlinear field dependence of electrophoresis can give rise to unexpected particle trapping in fluidic channels that is clearly different in origin from DEP trapping [105]. A high magnitude DC uniform electric field can induce nonlinear particle velocities, leading to particle flow reversal beyond a critical field magnitude, referred to as the electrokinetic (EK) equilibrium condition [106,107]. The concept of an amplification factor has been introduced to describe how an insulator constriction can greatly magnify the electric field—information that can be used to significantly reduce (to sub-100 V) the DC voltage required in an iDEP device to manipulate micron-sized particles [108]. This realization that in a so-called iDEP device the DEP force may in fact be largely irrelevant, with non-linear electrophoresis and electroosmosis effects the leading actors, has led to new particle separation strategies. An important example is provided by Quevedo et al. [109], who show that synthetic lysozyme and BSA sub-micron particles, as well as their blends, are separable based on differences in their isoelectric points and as manifested in large differences of the voltage required to selectively trap them. These results mirror to some extent the unique electrokinetic signatures found by Liu and Hayes for α-chymotrypsinogen, immunoglobulin G and lysozyme in their innovative gradient insulator-based (g-iDEP) device operated under DC conditions [23]. Although these proteins exhibited behaviours consistent with negative DEP, non-linear electrokinetic effects may have complicated this interpretation.

Until such time as electrothermally induced fluid flow and electroosmosis is proven to have not been responsible for the DEP cross-over at MHz frequencies shown for avidin, BSA and PSA in Figure 7, this effect should also be included in the “Something Else?” category.

## 6. Concluding Comments

Practitioners of DEP have backgrounds across the biological, chemical, engineering and physical sciences. Most will appreciate having a simple formula to use when predicting, modelling or analysing experimental data. In the standard DEP theory, for particles of known spheroidal size and shape and without a permanent dipole moment, [*CM*]_macroscopic_ serves this purpose very well. However, the only way to predict the DEP response of a solvated globular protein molecule in the frequency range 1 kHz to 100 MHz is to examine either its published dielectric *β*-dispersion or full spectrum (e.g., Figure 4 and Figure 5), or to obtain these details by means of dielectric spectroscopy and/or MD simulations.

A simple equation that can be derived from Equation (36) for a mixture of two polar molecules is of the form
(61)εmixtureω∝vpgkpmp21+ω2τp2+1−vpgkwmw21+ω2τw2
where *v_p_* is the volume fraction of the component of lowest concentration, with *v_p_* + *v_w_* = 1 where *v_w_* is the volume fraction of the other component. For the case of a small concentration of protein (*p*) dissolved in water (*w*), the dielectric spectrum should reveal two distinct and separable dispersions. The second term on the right-hand side of this equation represents the *γ*-dispersion due to relaxation of the water dipoles and is centred near 10 GHz. At ~250 MHz where $\left(1+\omega^{2}\tau_{w}^{2}\right)\approx 1$, and $\left(1+\omega^{2}\tau_{p}^{2}\right)\gg 1$, the decrement of the *γ*-dispersion should equal (1 − *v_p_*) and so provide an indication of the effective volume of the protein with its hydration shell. The magnitude and frequency dependence of the first term on the right-hand side of Equation (61) represents the *β*-dispersion and provides the information regarding the predicted DEP response (see Figure 10). For a protein of medium molecular weight (e.g., BSA), the small in amplitude but identifiable *δ*-dispersion near 70 MHz should be observed [41]. For proteins of low molecular weight (e.g., ubiquitin) the *δ*-dispersion manifests itself as a slight distortion of the high-frequency tail of the *β*-dispersion [75]. The combined magnitude of $g_{kp}m_{p}^{2}$ can be calculated or estimated from classical dielectric theory, but calculation of each component is only now, for the first time, possible as a result of the innovative theory and new microsecond MD simulations presented by Heyden and Matyushov [1].

Equation (9) provides a simple DEP force equation to be used for globular proteins, with the DEP susceptibility χ_DEP_ given by Equation (49). For lysozyme and ubiquitin, the magnitudes and frequency-dependencies of the susceptibility factor *Re*[*K*], obtained by Heyden and Matyushov [1], are shown in Figure 10. In the absence, for other proteins, of values for the susceptibility factor $\chi_{c}^{HM}$ of Equation (51) (which effectively replaces the empirical parameter, *b*, of Equation (48)) empirical values given elsewhere [4,24] can be used because they are based on experimental values for Δ*ε*(*β*)/*v_p_* and provide information regarding the relative magnitudes and frequency profiles to be expected. The new theory [1] and empirical one [4] do not take into account relaxations of electrical double layers and ion diffusion processes, and so cannot be relied upon to predict DEP responses for frequencies below ~1 kHz.

## Figures and Tables

**Figure 1 micromachines-13-00261-f001:**
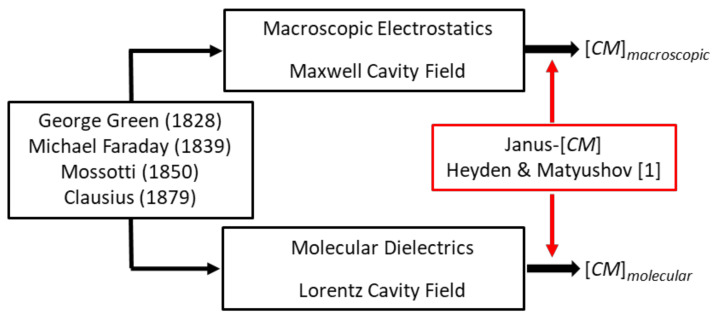
Standard DEP theory employs macroscopic electrostatics to calculate an induced dipole moment and the Clausius–Mossotti factor, [*CM*]*_macroscopic_*. The DEP of particles possessing a permanent dipole moment is better formulated within the context of the Clausius–Mossotti law of molecular dielectrics ([*CM*]*_molecular_*). The new theory [1] holds the key to transitioning between the two [*CM*]s, whose origins trace back to Green [7], Faraday [8], Mossotti [9] and Clausius [10].

**Figure 2 micromachines-13-00261-f002:**
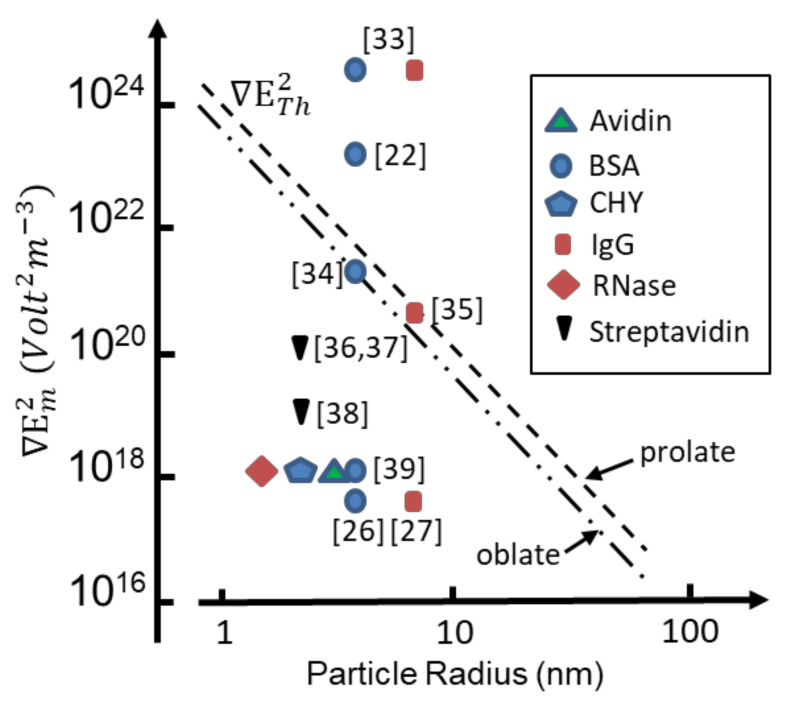
Reported values of $\nabla E_{m}^{2}$ employed in DEP experiments mapped against the effective (hydrodynamic) radius of the test protein molecule. The straight-line plots show the predicted threshold values of $\nabla E_{Th}^{2}$ required to overcome Brownian forces, derived using Equation (13) for prolate and oblate spheroids. (BSA: bovine serum albumin; CHY: chymotrypsinogen; IgG: immunoglobulin G.)

**Figure 3 micromachines-13-00261-f003:**
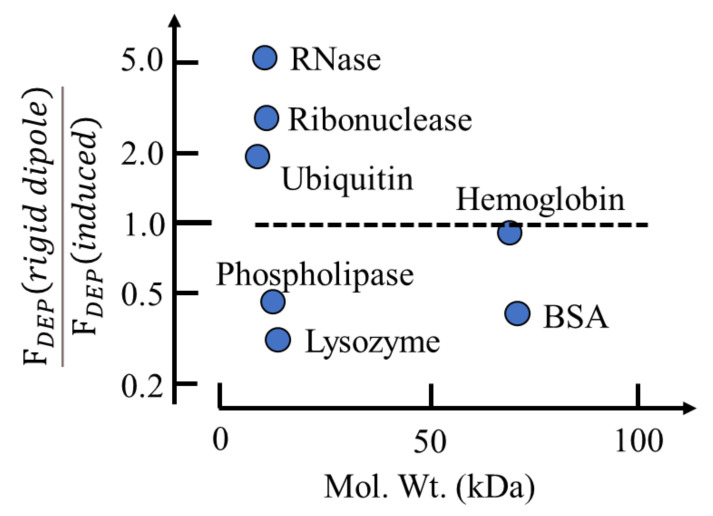
The contribution to the DEP force that a rigid dipole moment makes relative to that of the induced moment is shown for various proteins, as calculated using Equation (15). Values of the hydrodynamic radii were derived using an empirical relationship between protein size and molecular weight (Malvern Panalytical^®^—Zetasizer Nano ZS) and permanent dipole moment values were derived from the literature [4,24].

**Figure 4 micromachines-13-00261-f004:**
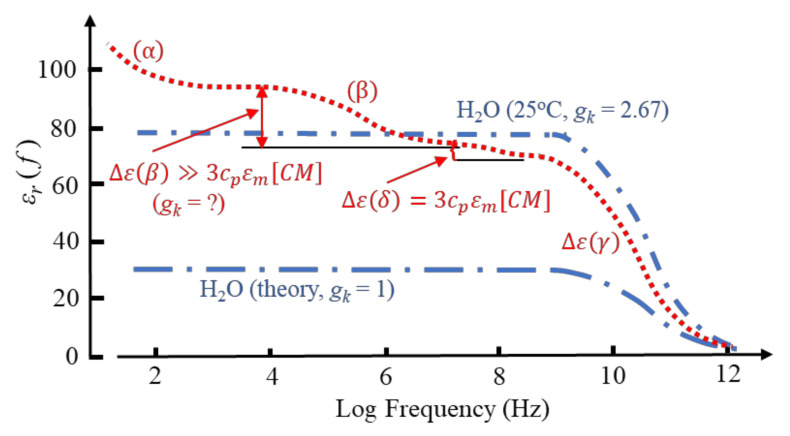
Protein solutions exhibit dielectric dispersions Δε, designated α, β, δ and γ, due to relaxations of the protein’s electrical double-layer; permanent dipole; hydration shell; bulk water dipoles, respectively [16]. The magnitude of Δε(β), unlike Δε(*δ*), greatly exceeds that predicted by Equation (16) [4]. Δε(γ) exhibited by pure water arises from correlations of its dipole moments (Kirkwood factor *g_k_* = 2.67), where *g_k_* = 1 corresponds to no correlation.

**Figure 6 micromachines-13-00261-f006:**
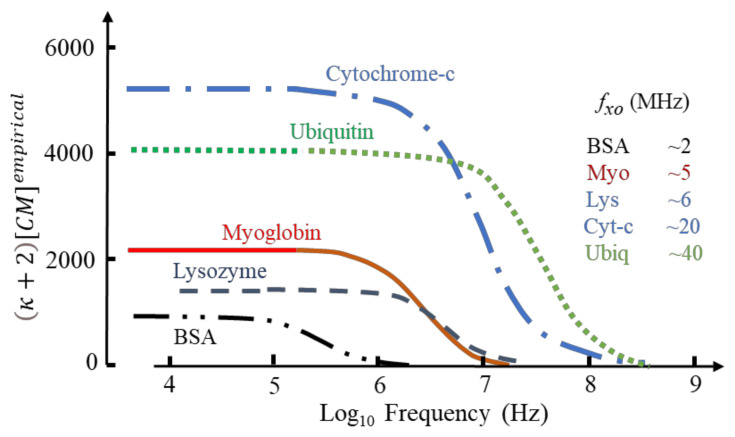
The frequency-dependence of the empirical factor $\left(\kappa +2\right){\left[CM\right]}^{empirical}$ derived from Equations (17) and (20) for selected proteins. The frequency, *f**_xo_*, marking the transition between the dielectric increment and decrement of the *β*-dispersion is also given.

**Figure 7 micromachines-13-00261-f007:**
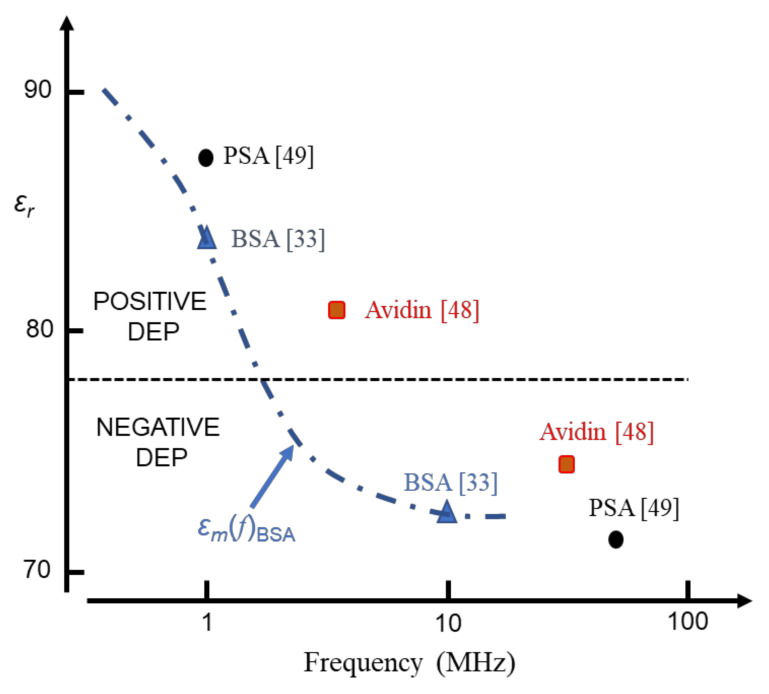
BSA, avidin and prostate specific antigen (PSA) have been reported to exhibit a DEP cross-over at MHz frequencies. This is consistent with the permittivity of the protein solution falling below the value *ε_r_* = 78 for pure water, as shown here for BSA (from data of Figure 6).

**Figure 10 micromachines-13-00261-f010:**
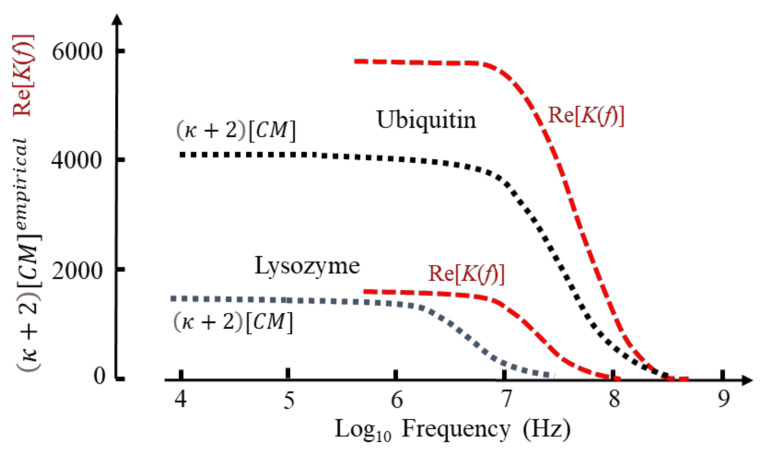
Frequency dependence of the DEP susceptibility factor *Re*[*K*] for ubiquitin and lysozyme [1], together with plots for their empirical factors (*κ* + 2)[*CM*] [4].

**Figure 11 micromachines-13-00261-f011:**
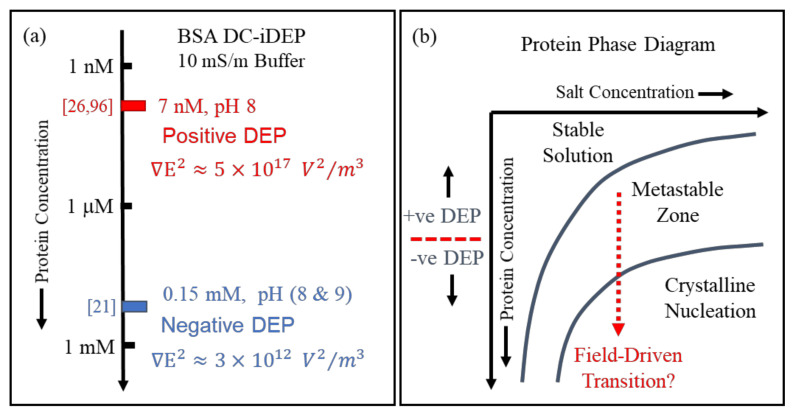
(**a**) BSA samples at the same buffer conductivity and pH exhibit negative iDEP at a concentration of 0.15 mM [21], but positive iDEP at 7 nM [26,98]. The low magnitude of $\nabla E_{m}^{2}$ associated with the negative DEP result suggests minimal influence of a non-linear electrokinetic or electrothermal effect. (**b**) A simple protein phase diagram [99,100] to show how crystallization might occur during the DEP of a protein sample in a metastable state.

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
