# Peer review of "Protein Dielectrophoresis: A Tale of Two Clausius-Mossottis—Or Something Else?"

_micromachines, 2022, doi:10.3390/mi13020261_

Round 1
Reviewer 1 Report
The referee has enjoyed reading this manuscript and wishes to thank the author for putting this rich body of theory together. A review like the present manuscript is of course relevant to the field and must be published. As the author carefully described in the last sections of the review, there are many areas of opportunity to further develop our knowledge of protein DEP. However, before recommending acceptance of the review, there are a few points that the referee considers important to discuss in greater detail.
All sections before section 5.2 dealt with the general AC stimulation voltage case. As a matter of fact, Matyushov’s papers model DEP phenomena for proteins starting at quite high frequencies. Three research groups (Alexandra Ros’s, Rafael Davalos’s and Nathan Swami’s) the referee is aware of have successfully implemented AC-iDEP and those works of Ros and Swami that deal with proteins have been correctly cited in the present manuscript. Davalos has not published works on protein DEP and therefore, there is no need to cite his papers.
In the DC case, however, DEP theories (either the conventional one or MD-based one) have not truly been able to provide an adequate interpretation of experimental observations. It is true that many papers were published under the DC-iDEP concept in the past 20 years. Nonetheless, Blanca Lapizco recently published a review paper highlighting the requirement of a quite large correction factor (sometimes as high as 600) to force the theoretical predictions resemble the experimental observations [A]. Shortly after the publication of that review, almost simultaneously, Ulrich Keyser’s group and the Perez-Gonzalez’s research group published a couple of works that demonstrated the importance of nonlinear electrophoresis in particle manipulation applications under large DC electric fields and the almost irrelevant presence of DEP in what had thus far been termed DC-iDEP [B,C]. The Lapizco-Encinas’s group followed this with a series of papers on the application of the new theory proposed in [C], characterizing the trapping of cells and protein nanoparticles [D-F]. In a parallel avenue, Xiangchun Xuan’s group has extensively explored the effects on nonlinear electroosmosis to particle trapping in the same type of devices [G]. Although only one of these papers deals with proteins (protein agglomerates to be fair), the referee considers that this should be mentioned in section 5.2 as a possible 3rd avenue to explain the experimental observations. The author has used one of the classic texts published by Blanca Lapizco [H] to exemplify a success-case of classic DEP theory in insulator-based devices, while Blanca Lapizco has recognized in her recent publications that in face of these new findings, those initial studies could not have been explained on the basis of the classic DEP theory [I]. Similar statements have been published by Perez-Gonzalez [J], Xuan [G], and Davalos [K]. As protein feature a net charge at certain pH values, it is very likely that nonlinear electrophoresis in combination with electroosmosis is responsible for particle trapping under DC conditions. Moreover, considering that Matyushov’s paper on protein DEP predicts positive DEP at “low frequencies” and no DEP at “high frequencies”, negative DEP seems unlikely to explain protein trapping in insulator-based devices as the author correctly indicated.
In addition to this, there are a few minor recommendations the author might want to consider.
- The referee recommends the author to use boldface non-italicized fonts for vectors and italicized non-boldface fonts for scalars in equations. Physicists and Electrical Engineers can easily follow the equations in this manuscript. However, it might not be the case for Chemists and Biotechnologists.
- Legend in Figure 2 shows the _ and ^ symbols instead of formatting the accompanying text as subscript and superscript, respectively.
- Line 935 “this these” should read “these”
References:
[A] Hill, Nicole, and Blanca H. Lapizco‐Encinas. "On the use of correction factors for the mathematical modeling of insulator based dielectrophoretic devices." Electrophoresis 40.18-19 (2019): 2541-2552.
[B] Tottori, Soichiro, et al. "Nonlinear electrophoresis of highly charged nonpolarizable particles." Physical review letters 123.1 (2019): 014502.
[C] Cardenas-Benitez, Braulio, et al. "Direct current electrokinetic particle trapping in insulator-based microfluidics: theory and experiments." Analytical Chemistry 92.19 (2020): 12871-12879.
[D] De Peña, Adriana Coll, et al. "Creation of an electrokinetic characterization library for the detection and identification of biological cells." Analytical and bioanalytical chemistry 412.16 (2020): 3935-3945.
[E] Antunez-Vela, Sofia, et al. "Simultaneous determination of linear and nonlinear electrophoretic mobilities of cells and microparticles." Analytical Chemistry 92.22 (2020): 14885-14891.
[F] Quevedo, Daniel F., et al. "Electrokinetic characterization of synthetic protein nanoparticles." Beilstein journal of nanotechnology 11.1 (2020): 1556-1567.
[G] Xuan, Xiangchun. "Review of nonlinear electrokinetic flows in insulator‐based dielectrophoresis: From induced charge to Joule heating effects." Electrophoresis 43.1-2 (2022): 167-189.
[H] Lapizco‐Encinas, Blanca H., et al. "Insulator‐based dielectrophoresis for the selective concentration and separation of live bacteria in water." Electrophoresis 25.10‐11 (2004): 1695-1704.
[I] Lapizco-Encinas, Blanca H. "The latest advances on nonlinear insulator-based electrokinetic microsystems under direct current and low-frequency alternating current fields: a review." Analytical and Bioanalytical Chemistry (2021): 1-21.
[J] Ruz-Cuen, Rodrigo, et al. "Amplification factor in DC insulator-based electrokinetic devices: a theoretical, numerical, and experimental approach to operation voltage reduction for particle trapping." Lab on a Chip 21.23 (2021): 4596-4607.
[K] Duncan, Josie L., and Rafael V. Davalos. "A review: Dielectrophoresis for characterizing and separating similar cell subpopulations based on bioelectric property changes due to disease progression and therapy assessment." Electrophoresis 42.23 (2021): 2423-2444.
Author Response
The author is extremely grateful to the reviewer for their careful reading of the manuscript, as well as for their excellent recommendations. Your review is amongst the very best I have ever received as an author! I have gained new knowledge as a result.
My responses take the form of new sentences (in red font) on pages 24 and 25, accompanied by new references (marked in red font) numbered 97, 98, 104-109. I hope you find these to be appropriate. I have also made the two small corrections in the main text that you have helpfully indicated. Thank you.
Reviewer 2 Report
This is another valuable addition from Dr. Pethig to the field of Dielectrophoresis. The inconsistency between predicted pDEP and nDEP properties for various protein molecules was for eDEP and iDEP-based applications. The recent molecular dynamics reports for different molecules have been discussed to explain some of these DEP behaviors observed over the past three decades. I also encourage the author to consider adding more recent DEP review articles (e.g. Anal. Chem. 2019, 91, 277−295 as it discusses various theoretical DEP considerations for molecules based on different shapes and shell structures). Overall, this is a valid and much-needed discussion in the field of DEP, and I strongly suggest accepting this manuscript.
Author Response
The author thanks the reviewer for their positive feedback and the valid suggestion that more recent review articles should be included. I have added such reviews as new references 90, 91, 104 and 106, marked in red font. Sentences (in red font) have been added on pages 23/24 and 25 to accompany them. I hope the reviewer considers this to be adequate and appropriate. Thank you.
Reviewer 3 Report
Author suggests a new approach for determination of protein dielectrophoresis in the perspective of eDEP and iDEP conditions.
I think this manuscript is enough to be accepted and published in present form in Micromachine journal.
Author Response
The author thanks the reviewer for their positive recommendation.